# Coastal primary productivity changes over the last millennium: a case study from the Skagerrak (North Sea).

Anna Binczewska[1], Bjørg Risebrobakken[2], Irina Polovodova Asteman[2,5], Matthias Moros[3], Amandine Tisserand[2], Eystein Jansen[2,4], Andrzej Witkowski[1]

5   [1] Faculty of Geosciences, University of Szczecin, Szczecin, Poland
    [2] Uni Research Climate, Bjerknes Centre for Climate Research, Bergen, Norway
    [3] Leibniz Institute for Baltic Sea Research (IOW), Warnemünde, Germany
    [4] Department of Earth Science, University of Bergen, Norway
    [5] Currently at: Marin Mätteknik (MMT) Sweden AB, Gothenburg, Sweden

10  *Correspondence to*: Anna Binczewska (anna.binczewska@usz.edu.pl)

**Abstract.** A comprehensive multi-proxy study on two sediment cores from western and central Skagerrak was performed in order to detect the variability and causes of marine primary productivity changes in the investigated region over the last 1100 years. The cores were dated by Hg pollution records and AMS $^{14}$C dating and analysed for palaeoproductivity proxies such as total organic carbon, $\delta^{13}$C, total planktonic foraminifera, benthic foraminifera (total assemblages as well as abundance of *Brizalina skagerrakensis* and other palaeoproductivity taxa) and palaeothermometers such as Mg/Ca and $\delta^{18}$O. Our results reveal two periods with changes in productivity in the Skagerrak region: i) a moderate productivity at ~ CE 900 – 1700 and ii) a high productivity at ~ CE 1700 – present. During ~ CE 900 – 1700, moderate productivity was likely driven by the nutrients transported with the warm Atlantic water inflow associated with a tendency for a persistent positive NAO phase during the warm climate of the Medieval Climate Anomaly, which continues into the LIA until ~ CE 1450. The following lower and more variable temperature period at ~ CE 1450 – 1700 was likely caused by a reduced contribution of warm Atlantic water, but stronger deep-water renewal, due to a generally more negative NAO phase and a shift to the more variable and generally cooler climate conditions of the Little Ice Age. The productivity and fluxes of organic matter to the seafloor did not correspond to the temperature and salinity changes recorded in the benthic *Melonis barleeanus* shells. Since ~ CE 1700 towards present day our data point to an increased nutrient content in the Skagerrak waters. This increased nutrient content was likely caused by enhanced inflow of warm Atlantic water, increased Baltic outflow, intensified river runoff and enhanced human impact through agriculture expansion and industrial development. Intensified human impact likely increased nutrient transport to the Skagerrak and caused changes in the oceanic carbon isotope budget, known as the Suess effect, which is clearly visible in our records as a negative shift in $\delta^{13}$C values from ~ CE 1800. In addition, a high appearance of *S. fusiformis* during the last 70 years at both studied locations suggests increased decaying organic matter at the sea floor after episodes of enhanced primary production.

## 1 Introduction

Growth of marine microalgae is stimulated by enhanced concentrations of major nutrients like nitrogen and phosphorus in the photic zone (e.g. Sigman and Hain, 2012). Microalgae in the oceans are primary producers, which provide food for consumers at higher trophic levels and oxygen for respiration (e.g. Micheli, 1999). Through photosynthesis and the biological pump, marine primary producers also extract $CO_2$ from the atmosphere. Carbon, nutrients and trace elements, which are fixed by primary producers are further ingested by higher organisms or sink to the ocean floor where they are stored in the form of organic matter. The organic matter will eventually be remineralised, releasing the carbon, nutrients and trace elements to the bottom water (e.g. Sigman and Hain, 2012). A fraction is also buried in sediments. Supply of carbon to the oceanic ecosystems is an important part of the biogeochemical cycles, which presently are being disturbed by the human impact. Via photosynthesis, primary producers can help to remove $CO_2$ from the atmosphere through the biological pump. However, increasing levels of dissolved $CO_2$ in the oceans associated with so-called "ocean acidification" may negatively impact ocean's carbonate producers (e.g. Doney et al., 2009; Haynert et al., 2012) by influencing their survival and fitness (e.g. Thomsen et

al., 2017). Excessive export of organic matter furthermore changes the oxygen condition at the sea floor due to decay of organic matter, which lowers the dissolved oxygen content, in turn, negatively influencing the benthic life regime (e.g. Kristiansen and Aas, 2015 and references therein).

Coastal zones are among the most productive marine regions, characterised by high atmospheric $CO_2$ uptake, organic matter accumulation and decomposition (e.g. Hjalmarsson et al., 2010). The Skagerrak, located between the North Sea and the Baltic Sea and in the close proximity to land, has many potential nutrient sources, such as the North Atlantic, Baltic Sea, North Sea, as well as continental discharge and river runoff (Aure and Dahl, 1994; Andersson, 1996; Gustafsson and Stigebrandt, 1996). Upwelling and precipitation further increase the nutrient supply to the surface waters, additionally stimulating productivity in this region (Pingree et al., 1982; Aure and Dahl, 1994; Fonselius, 1996). The North Sea and the Skagerrak absorb significant quantities of atmospheric $CO_2$ via the biological pump (1.38 mol C $m^{-2}$ $yr^{-1}$ and 1.2 mol C $m^{-2}$ $yr^{-1}$, respectively) and thus play an important role in the carbon cycle (Thomas et al., 2005; Hjalmarsson et al., 2010). The Skagerrak acts as a main depositional basin for about half of the refractory carbon produced in the North Sea and for a high amount of labile organic matter either imported with waters of the near-bottom current from the Danish coast or produced by intense algal blooms (Boon et al., 1999). Input of nutrients largely regulate food webs, which makes nutrients of great economic importance for the coastal areas worldwide (Micheli, 1999; FAO, 2016). In the Skagerrak nutrients are particularly important for the Nordic fisheries (Hop et al., 1992; Iversen et al., 2002; Olsen et al., 2004; Skogen et al., 2007). Fisheries and aquaculture sectors of the Skagerrak, commercially valuable for the Scandinavian nations, also make a relevant contribution to a growing global demand for food, which will largely rely on coastal regions to host a major part of food production in the future (e.g. FAO, 2013). Effects of increased primary production range from positive impacts on growth rate, size and reproduction of fish and shellfish populations to disruptive alterations in the food webs, thus yielding or reducing the profit rates of fisheries (Hop et al., 1992; Micheli, 1999; Iversen et al., 2002; Olsen et al., 2004, 2005; Breitburg et al., 2009; FAO, 2016). Disruptive changes in trophic levels of the Skagerrak ecosystem have been attributed to overfishing (Cardinale and Svedäng, 2004) but ongoing studies alert about adverse impact of increased nutrient inputs driving heavy phytoplankton blooms and eutrophication in the region (e.g. Baden et al., 1990; Aure et al., 1996; Breitburg et al., 2009). Eutrophication causes high demand and depletion of oxygen in the bottom waters, which affect species diversity, morphology and population growth, and forces organisms to migrate (Rosenberg et al., 1990; Conley, 2009). Thus, to better understand the ongoing and possible future productivity changes and associated environmental effects, more historical studies are needed.

Although the Skagerrak is a well-investigated area, retrospective studies from this region have focused on climate change instead of productivity, and hence, past primary productivity changes are not well known (Polovodova Asteman et al., 2018). Previous studies suggested that among the processes driving primary productivity changes in the Skagerrak are 1) the North Atlantic Oscillation (NAO), 2) an input of anthropogenic nutrients from the Skagerrak catchment area and 3) nutrients transported from the Baltic Sea. Thus, while the NAO influences the inflow of the nutrient-rich Atlantic water mass into the Skagerrak (Gil et al., 2005; Brückner and Mackensen 2008), anthropogenic activities in the Skagerrak catchment area, such as land-use changes, also cause an increased nutrient input and high organic matter flux to the basin (e.g. Filipsson and Nordberg,

2010). In addition, eutrophication of the Baltic Sea and resulting transport of nutrient-rich water with the Baltic Current through surface water exchange processes represents potentially an additional nutrient source (Andersson, 1996; Hjalmarsson et al., 2010; Filipsson et al., 2017; Krossa et al., 2015; Polovodova Asteman et al., 2018). In its turn, a nutrient-overloaded ecosystem in the Skagerrak may further supply dissolved inorganic nitrogen (DIN) to the Kattegat bottom waters (Carstensen et al., 2006) and increase the nutrient level in the coastal waters of Norway (Rosenberg et al., 1987; Rydberg et al., 2006) (Fig. 1). The last two processes involve human induced influences on the ecosystem, while the NAO relates to effects following variability in the climate system. Over the last 1100 years the interaction between these processes has changed, as human influence has increased. In perspective of the last 4500 years, it is seen that productivity and its variability has increased from 50 years before the Common Era (BCE) towards present day (Polovodova Asteman et al., 2018). The increased productivity occurred as a response to enhanced local runoff coinciding with high winter rainfall and general cooling in Scandinavia, as well as intensified Baltic outflow, which may have significantly contributed to the nutrient supply in the Skagerrak in the past (Polovodova Asteman et al., 2018).

Our study aims to detect the variability and causes of marine primary productivity changes in the Skagerrak over the last 1100 years. We address this aim through a comprehensive multi-proxy study of two sediment cores from the central and the western Skagerrak, integrating records of the total organic carbon (TOC), foraminiferal assemblage data, stable carbon and oxygen isotopes ($\delta^{13}$C, $\delta^{18}$O) and trace element ratio (Mg/Ca). In addition, we evaluate our results in context of the long-term productivity changes described previously in Polovodova Asteman et al. (2018).

## 2 Study area

The Skagerrak basin is located in the northeastern part of the North Sea, connected to the Baltic Sea through the Kattegat (Fig. 1). The basin has a mean water depth of 210 m and a sill depth of 270 m. With a maximum depth of 700 m, the Skagerrak represents the deepest part of the Norwegian Trench (Rodhe, 1996). The area is characterised by an anticlockwise circulation and complex hydrography. The surface circulation (<30 m) is to a large extent dominated by a surface current consisting of inflowing saline water from the southern North Sea and the North Atlantic and outflowing less saline water from the Baltic Sea (Danielssen et al., 1997). The inflowing nutrient-rich surface water flows along the Danish coast driven by the Southern Jutland Current (SJC) and the Northern Jutland Current (NJC) while the outflowing Baltic Sea water (BW ~20 – 30 psu) flows as the Baltic Current (BC) along the Swedish west coast towards the northeast Skagerrak where it merges with the NJC and turns to the northwest as the low saline Norwegian Coastal Current (NCC) (Rodhe, 1996; Rydberg et al., 1996). The water flowing from the Skagerrak towards the Norwegian Sea (NCC) partly recirculates to the western Skagerrak (Rodhe, 1996). The surface water has a high nutrient concentration mostly due to the freshwater input via rivers draining from the Norwegian south coast, German and Danish east coasts, and the Baltic catchment area but also the upwelling of the underlying nutrient-rich Atlantic water is considered to be an additional nutrient supply (Gustafsson and Stigebrandt, 1996; Rodhe, 1996). As a consequence of the mixing of different water types and the high freshwater input enhanced by precipitation, the upper layer of the surface water has low salinity (25 – 32 psu) and is determined as the Skagerrak Coastal Water (4.5 – 10 °C). The

intermediate water layer (30 – 270 m) is referred to as Skagerrak Water (32 – 35 psu, 4.5 – 10 °C) and is driven by the subsurface circulation (Andersson, 1996). The deep-water layer below sill depth (>270 m) is dominated by Atlantic Water and is recognised as the Skagerrak Basin Water (>35 psu, 5.5 – 6.5 °C) (Aure and Dahl, 1994).

The subsurface circulation (below 30 m water depth), consists of nutrient-rich Atlantic deep water (AW >35 psu, 5.5 – 8.5 °C)
flowing through the northern North Sea, and the water from the central and southern North Sea (NSW ~31 – 35 psu) (Rodhe, 1996). The inflowing water follows the southern side of the Norwegian Trench, entering the Skagerrak in its central part where this water is mixed with fresh surface-water and flows out as the NCC (Winther and Johannessen, 2006).

Large-scale atmospheric systems and regional meteorological factors (e.g. precipitation and storms) influence the flow regime creating a high-dynamic system in the upper layer of the water column where water mixing is largely caused by the
southwesterly winds (Gustafsson and Stigebrandt, 1996). At the same time, calmer hydrographic conditions are typical for the intermediate layer and the deep water down to ~ 400 m with a maximum water residence time of 3 months (Andersson, 1996). This is in contrast to the renewal of the deepest water mass below ~ 400 m, which is replenished every 1 to 3 years depending on the strength of the Atlantic water inflows (Aure and Dahl, 1994; Rodhe 1996), closely correlating with the NAO index (Brückner and Mackensen, 2006).

**3 Material and methods**

Two gravity cores (GC) were retrieved from the Skagerrak during a R/V Elisabeth Mann-Borgese cruise in May 2013. Core EMB046/20-3GC (4.8 m long) was taken from the central Skagerrak (SE Norwegian Trench; 58°31.75´N, 09°29.13´E; 533 m water depth), while core EMB046/10-4GC (4.62 m) comes from the western Skagerrak (SW Norwegian Trench; 57°49.73´N,
07°17.62´E; 457 m water depth) (Fig. 1). Both cores were taken below sill depth within the deep waters of the Norwegian Trench. The cores were cut in 1-m sections on board before being split and subsampled every 1 cm ashore. This study is based on results from the upper 170.5 and 164.5 cm of cores EMB046/20-3GC and EMB046/10-4GC respectively, which corresponds to the last 1100 years. A part of the foraminiferal dataset and the TOC data were previously published in Polovodova Asteman et al. (2018). Here we present new stable isotopes ($\delta^{13}$C, $\delta^{18}$O), and trace element ratio (Mg/Ca) data
covering the last 1100 years, in combination with foraminiferal assemblage data and multivariate statistics. Both cores consist of mostly homogeneous soft organic-rich clay, have olive-grey colour and show no significant changes in grain size and lithology throughout the studied intervals. The TOC content was determined using 'Rapid CS cube –Elementar' analyser (Department of Geology and Paleogeography, University of Szczecin, Poland) with a measurement accuracy of 0.01% (95% confidence level (CL) at 99.5% detection limit (DL)). For detailed methodology of geochemistry measurements (TOC) see
Polovodova Asteman et al. (2018).

For the stable carbon and oxygen isotopes ($\delta^{13}$C, $\delta^{18}$O) and trace element analyses, well-preserved tests of benthic foraminifera *Melonis barleeanus* were picked from the dried sediment (fraction >150 µm). *Melonis barleeanus* was selected for the analyses due to its relatively high abundance throughout the investigated intervals at both sites and its well-known

potential for geochemical palaeoreconstructions (Mackensen et al., 2000; Kristjánsdóttir et al., 2007; Brückner and Mackensen, 2008; Butruille et al., 2017). Stable isotope measurements were performed at 1 cm intervals and the trace elements analyses were done at $1-2$ cm intervals from both gravity cores down to 170 cm (EMB046/20-3GC) and 164.5 cm (EMB046/10-4GC). No stable isotope or trace element analyses were done between 4.5 cm and 7.5 cm in EMB046/10-4GC, due to lack of material, because most of the foraminifera from that interval where used for $^{14}C$ AMS dates.

Stable isotope analyses were run on a Finnigan MAT 253 mass spectrometer equipped with an automatic Kiel device at FARLAB of the University of Bergen. Prior to measurement the tests of *M. barleeanus* were lightly crushed, cleaned with methanol ($\geq$ 98.8 %) using an ultrasonic bath and dried at 60 ºC. For each measurement $2-4$ specimens were used. All results are reported in ‰ versus Vienna Pee Dee Belemnite standard (V-PDB), using the National Bureau of Standards (NBS) 19 and 18, in combination with the internal lab standard CM12. The long-term analytical uncertainty is ±0.08‰ and ±0.03‰ (95% CL) for oxygen and carbon isotopes, respectively.

Shells of *M. barleeanus* were also cleaned and analysed for trace elements at the Trace Element Lab (TELab) at Uni Research Climate, Bergen (Norway). For each analysis, approximately 15-20 specimens were gently crushed between two glass plates under a microscope to allow the contaminants to be removed. The samples were cleaned following the procedure described by Barker et al. (2003). The cleaning method includes clay removal steps, oxidation of the organic matter and surface leaching. Samples containing enough material were mixed and split into two subsamples to allow duplicate analysis. All samples were dissolved in trace metal pure 0.1M $HNO_3$ (prepared from $HNO_3$ TraceSELECT®) and diluted to a final concentration of 40 ppm of calcium (Ca).

The trace elements were measured on an Agilent 720 inductively coupled plasma optical emission spectrometer (ICP-OES). Six standards have been prepared at the TELab and they have a composition similar to foraminiferal carbonate (0.50 – 7.66 mmol mol$^{-1}$). Fe/Ca and Al/Ca values have been checked and showed no significant correlation with the measured Mg/Ca values. The correlation coefficients ($R^2$) between Mg/Ca and, Fe/Ca and Al/Ca ratios are for the core EMB046/20-3GC, 0.012 and 0.095, and for the core EMB046/10-4GC, 0.020 and 0.067, indicating no systematic contamination due to insufficient cleaning. The Mn/Ca values in our samples are higher than the recommended maximum (<105 μmol mol$^{-1}$) (Boyle, 1983), indicating that diagenetic coatings might also affect our results. The Mn/Ca values, however, show no significant correlation with the measured Mg/Ca values ($R^2 = 0$ for EMB046/20-3GC and $R^2 = 0.021$ for EMB046/10-4GC). Standard solution with Mg/Ca of 5.076 mmol mol$^{-1}$ is analysed after every eight samples to correct for instrumental biases and analytical drift of the instrument. The long-term Mg/Ca analytical precision, based on the standard solution, is ±0.016 mmol mol$^{-1}$ (1σ Standard Deviation) or 3.11% (relative SD). Average reproducibility of duplicate measurements (pooled SD, dof = 41) is equivalent to an overall average precision of 4.09%. The average Mg/Ca of long-term international limestone standard (ECRM752-1) measurements was 3.76 mmol mol$^{-1}$ (1σ = 0.07 mmol mol$^{-1}$) with the average published value of 3.75 mmol mol$^{-1}$ (Greaves et al., 2008).

The *Melonis* spp. Mg/Ca - Bottom Water Temperature (BWT) is calculated from the measured Mg/Ca using the new *Melonis* spp. calibration (Mg/Ca = 0.113(±0.005)*BWT + 0.792(±0.036)), based on core top data covering a Mg/Ca range of

0.68 – 3.66 mmol mol$^{-1}$ for a temperature range of -0.89 – 15.58°C (Hasenfratz et al., 2017). According to the calibration uncertainty, a 1σ temperature error (95% CL) of ±0.9°C to ±1.7°C has to be taken into consideration for the temperature range (4.1 – 9°C) covered by EMB046/20-3GC. For EMB046/10-4GC the 1σ temperature error (95% CL) is similar and ranges between ±1°C to ±1.6°C for the temperatures 5°C – 8.5°C. Further discussion on Mg/Ca-derived BWT within this article will

utilize the Hasenfratz's calibration (Hasenfratz et al., 2017).

The results from Mg/Ca-derived BWT and δ$^{18}$O were used to calculate the sea water isotopic composition (δw) using the following equation: $\delta w = ((\delta^{18}O_{top} - \delta^{18}O_{down}) - (T_{top} - T_{down})*0.23)+0.3$. The $\delta^{18}O_{top}$ is the δ$^{18}$O value measured on the foraminiferal shells from the upper most sample (0-1 cm core depth) and the $T_{top}$ is the temperature taken from CTD measured at time of coring. The $\delta^{18}O_{top}$ equals 2.04 and 2.06 (‰) while $T_{top}$ was 5.72 and 5.34°C, respectively for the EMB046/10-4GC

and EMB046/20-3GC. The $\delta^{18}O_{down}$ and $T_{down}$ is the down core δ$^{18}$O and temperature as measured on the foraminifera samples. Temperature estimates based on δ$^{18}$O follows Shackleton (1974), where 0.23‰ equals 1°C in the temperature interval estimated for these sites. A constant of 0.3‰ is used to correct for the difference between VPDB and δ$_w$. No correction for an ice volume effect was applied as this is considered negligible over the last 1000 years.

The foraminiferal analysis was carried out on 5 – 10 g wet sediment, gently sieved over a 63-μm sieve and wet-counted

for foraminifera immediately afterwards. The two Skagerrak records were counted, over the targeted time interval covering the last 1100 years, at 1 – 3 cm resolution with an exception of 7 cm interval between 89 cm and 96 cm (~ CE 1030 – 1055) for EMB046/20-3GC, and at 1 – 2 cm resolution for the EMB046/10-4GC record. In the >63 μm fraction, at least 300 benthic and 300 planktonic (where possible) specimens were counted under a stereomicroscope and identified to a species level. Both relative (%) and absolute (individuals per gram in wet sediments, as ind. g$^{-1}$wet sed.) abundances were calculated. The benthic

foraminiferal species were categorised depending on their relative abundance in the assemblage to dominant (>10%), accessory (5% – 10%) or rare (<5%). Only dominant and accessory species are discussed in this study (Fig. 3, Table 2). Benthic species with a relative abundance of >5% in at least two samples were subject to multivariate statistics using simple CABFAC factor analysis with varimax rotation (Table 2), performed by the PAST software (Hammer et al., 2001). This statistical tool provides a reliable method to distinguish the statistically significant foraminiferal units dominated by different species (e.g. Polovodova

Asteman et al., 2013). In addition, benthic foraminiferal species indicative of increased organic matter fluxes to the sea floor and, hence, algal blooms, were grouped as '*palaeoproductivity fauna*' and included: *Alabaminella weddelensis, Brizalina skagerrakensis, Bulimina marginata, Epistominella* spp.*, Nonionella iridea* and *Uvigerina* spp. (Polovodova Asteman et al., 2018 and references therein). The planktonic foraminifera are presented as total planktonic individuals.

**4 Chronology**

A common age model has previously been established for the two cores (Polovodova Asteman et al., 2018). The two age models were set at a common depth scale based on 30 available AMS $^{14}$C dates, as well as a correlation between total inorganic carbon, relative abundance of *B. skagerrakensis* and mercury (Hg) records from both cores. All dates were calibrated using

Calib 7.10 (Stuiver et al., 2017), Marine13 (Reimer et al., 2013) and ΔR=0±50. The age model of Polovodova Asteman et al., (2018) is reasonable when investigating the longer-term trends over the last 4.5 ka. However, when focusing on the last 1100 years we found that there was a need for an improvement of the age model over this time period. This was achieved by increasing the number of dates and by a fine-tuning of the reservoir age. The new age model of EMB046/20-3GC, reaching back to CE 295, is based on 8 [14]C AMS dates in addition to the initial Hg increase at CE 1900 (Moros et al., 2017). The core top was set to 2013, the year of coring. A modern core top age is confirmed by a post bomb age at 5.5 cm, as well as a recording of the Cs-137 signal associated with the nuclear weapons testing period (not shown). Considering all of the information from the upper part of the core in detail makes it clear that the use of a ΔR=200±50 provides a better transfer, avoiding an unlikely jump in sedimentation rate, between the modern ages and the [14]C ages than when using a reservoir age of ΔR=0±50. Hence, when establishing the new age model, the [14]C AMS dates were calibrated using Calib 7.10 (Stuiver et al., 2017), Marine13 (Reimer et al., 2013) and ΔR=200±50, and linear interpolation between the established tie points (Table 1; Fig. 2). The new ages calculated for EMB046/10-4GC are based on the new age model of EMB046/20-3GC and the previously established common depth scale for EMB046/10-4GC and EMB046/20-3GC (Polovodova Asteman et al., 2018) (Fig. 2). Due to the established relationship between the depth scales of the two cores (Polovodova Asteman et al., 2018), the age model for EMB046/20-3GC can be, and are, used to create the new age model for EMB046/10-4GC.

## 5 Results

### 5.1 Organic geochemistry of bulk sediment

Both records show low TOC values until ~ CE 1700, around 1.7 – 2.1% and 1.5 – 1.8% in EMB046/20-3GC and EMB046/10-4GC, respectively (Fig. 5). From ~ CE 1700, the TOC content strongly increases towards the core tops, where ranges 1.85 – 2.5% (EMB046/20-3GC) and 1.75 – 2.3 (EMB046/10-4GC) are recorded. When comparing the two cores, the TOC values are higher for the EMB046/20-3GC record than for EMB046/10-3GC (Fig. 5).

### 5.2 Carbon isotopes

Both δ[13]C records show similar long-term variations through the study interval. Between CE 1500 and 1700 there is, however, a distinct increase of δ[13]C values for EMB046/20-3GC and a decrease for EMB046/10-4GC. Mean δ[13]C values of the time interval between ~ CE 900 and 1700 are generally higher (-0.53‰ on average in EMB046/10-4GC and -0.44‰ on average in EMB046/20-3GC), than during ~ CE 1700 – 2000 when mean δ[13]C values of -0.73‰ (EMB046/10-4GC) and -0.58‰ (EMB046/20-3GC) are observed (Fig. 5). From ~ CE 1700 towards the present, both records show a strong decreasing δ[13]C trend from ca. -0.2‰ to -1.6‰, where generally lower absolute δ[13]C values are recorded in EMB046/10-4GC than in EMB046/20-3GC (Fig. 5).

### 5.3 Oxygen isotopes

Both Skagerrak records display similar $\delta^{18}O$ values, ranging from around 1.7 to 2.7‰ (Fig. 4). In general, the $\delta^{18}O$ in EMB046/20-3GC shows lower values at ~ CE 1050 – 1350, somewhat higher or more variable values between ~ CE 1400 and 1550, followed by a decrease until ~ CE 1700. The overall lower $\delta^{18}O$ values in EMB046/10-4GC between ~ CE 900 and 1550, characterised by lowest recorded $\delta^{18}O$ at ~ CE 1350 – 1550, is interrupted by an increase at ~ CE 1200 – 1350. Consequently, the $\delta^{18}O$ long-term trend is not common for both records, however there is one distinct period in both records of relatively high $\delta^{18}O$ values between ~ CE 1700 and 1800, after which the $\delta^{18}O$ gradually decrease until ~ CE 1850, and followed by a steady increased $\delta^{18}O$ in EMB046/20-3CG and more variable but overall lower $\delta^{18}O$ values in EMB046/10-4GC towards the core top (Figs. 4, 5).

## 5.4 Mg/Ca analyses and BWT

The Mg/Ca values vary from 1.33 to 1.87 mmol mol$^{-1}$ in EMB046/20-3GC record and from 1.37 to 1.97 in EMB046/10-4GC, in general giving an estimated BWT range between 4.7 and 8.1 ºC, which is within the range of instrumentally recorded temperatures (ICES, 2010; Fig. 4B). This comparison and further interpretation refer to the smoothed data, while the raw data are mostly within the range of the instrumental data, but not completely. Through the records, there is in general a good correlation between the BWT changes and the variability of the $\delta^{18}O$ values, showing similar patterns with periods of higher BWT corresponding to those with decreased $\delta^{18}O$ values and vice versa, except between ~ CE 1000 and 1150 in EMB046/20-3GC where both proxies show relatively high values (Fig. 4). Hence, there is consistency between the proxies within each core but not between the cores (Figs. 4, 5). Overall, BWT in EMB046/10-4GC is characterised by relatively little variability between ~ CE 900 and 1550 followed by higher variability in the records. In contrast, BWT in EMB046/20-3GC shows first decreasing trend until ~ CE 1500, somewhat higher but variable values between ~ CE 1550 and ~ CE 1700, drop in values at ~ CE 1700 – 1800 and again warmer BWT are shown for the youngest part of the record (~ CE 1800 – 2000) (Figs. 4, 5).

## 5.5 Water isotopic composition

The changes in δw records follow the pattern in the BWTs curves in both records. The decrease in δw values corresponds to periods of low temperature and vice versa. Lower correlation between δw and BWT was observed in the early part of the EMB046/20-3GC record (~ CE 1000 – 1150) (Figs. 4, 5).

## 5.6 Foraminiferal assemblages

Eight planktonic foraminiferal species are identified in both records. *Globigerinita uvula* and *G. glutinata* are the most abundant species, while *Globorotalia inflata*, *Globigerina bulloides*, *Neogloboquadrina pachyderma*, *Neogloboquadrina incompta*, *Turborotalia quinqueloba* and *Orbulina universa* are less abundant. However, in this study all planktonic species are presented together as total individuals per gram of wet sediments (Fig. 3) due to their overall low absolute abundance, varying between 0 and 49 ind. g$^{-1}$ wet sediment (EMB046/20-3GC) and 1.6 – 110 ind. g$^{-1}$ wet sediment (EMB046/10-4GC).

The planktonic foraminifera are most abundant in the interval between ~ CE 900 and 1700 in EMB046/10-4GC and at ~ CE 900 – 1550 in EMB046/20-3GC, after which they decrease towards the top of the cores and almost disappear after ~ CE 1850 in both records (Figs. 3, 5).

The benthic foraminiferal record from core EMB046/10-4GC is characterised by consistently high absolute abundances (123 – 455 ind. g$^{-1}$ wet sed.) in contrast to overall lower values in core EMB046/20-3GC (43 – 527 ind. g$^{-1}$ wet sed., where the highest value represents an individual peak above 361 ind. g$^{-1}$ wet sed. significantly standing out from the rest of the record). In EMB046/20-3GC, the absolute abundance of benthic foraminifera is high until ~ CE 1400. At CE 1400, then values drop below 150 ind. g$^{-1}$ wet sediment for the next ~ 300 years. After ~ CE 1700 the absolute abundances gradually increase to reach the highest recorded values between CE 1800 and 1900. A similar absolute abundance trend is shown for agglutinated foraminifera, however, those appear in higher numbers in core EMB046/10-4GC than in EMB046/20-3GC (Fig. 3).

The benthic foraminiferal assemblages consist of up to 61 and 57 species in the cores EMB046/20-3GC and EMB046/10-4GC, respectively. Among those, eight species are dominant (>10%) and nine are accessory (5-10%) in core EMB046/20-3GC while the EMB046/10-4GC record has six dominant and five accessory taxa (Table 2). The common dominant species for both cores include *Brizalina skagerrakensis*, *Cassidulina laevigata*, *Eggereloides medius*, *Nonionella iridea*, *Pullenia osloensis* and *Stainforthia fusiformis* (for a full list of dominant and accessory species see Table 2). Among benthic foraminiferal species *Brizalina skagerrakensis* shows the most prominent and consistent changes when comparing both records (Fig. 3).

The CABFAC factor analysis distinguished three factors for each of the cores (Fig. 3), which together explain 95% (EMB046/20-3GC) and 97% (EMB046/10-4GC) of the total variance (Table 3). The foraminiferal species with absolute value of factor scores >1 are considered to contribute significantly to the defined foraminiferal assemblages (Table 4) and are used to name the distinguished factors (assemblages). Thus, "*Pullenia osloensis* assemblage" associated with Factor 1 explains 81% (EMB046/20-3GC) and 86% (EMB046/10-4GC) of the variance, and includes species *P. osloensis* and *Nonionella iridea* defined for both records with an addition of *Cassidulina laevigata* in the EMB046/10-4GC dataset. The "*Brizallina skagerrakensis* assemblage" associated with Factor 2 is dominated by species *B. skagerrakensis* and explains ~11% (EMB046/20-3GC) and ~9% (EMB046/10-4GC) of the variance. Finally, factor 3 explains ~3% (EMB046/20-3GC) and ~1.8% (EMB046/10-4GC) of the variance and includes *N. iridea* as a common species for both records, with *P. osloensis* as the second dominant species for the EMB046/20-3GC record and *Stainforthia fusiformis* for the EMB046/10-4GC record, consequently resulting in "*N. iridea - P. osloensis*" (EMB046/20-3GC) and "*N. iridea - S. fusiformis*" (EMB046/10-4GC) assemblages (Table 3). The individual factor weights (importance) for each counted sample are expressed by factor loading (Fig. 3). Factors with loadings above 0.5 are considered most significant. The factor analysis shows that both records are defined by a clear dominance of the *P. osloensis* factor alternating with the *B. skagerrakensis* factor between CE 900 and 1700. The most pronounced changes in the foraminiferal assemblages occur between CE 1700 and the present day where the *P. osloensis* factor is to a large extent replaced by the *B. skagerrakensis* factor. Similar long-term variability is seen in the '*palaeoproductivity fauna*' group due to a strong dominance of *B. skagerrakensis* in this group (Fig. 3). In addition,

*palaeoproductivity fauna* appears in higher abundance between ~ CE 900 and 1200. In the uppermost part (~ CE 1950) of the EMB046/10-4GC record the *N. iridea- S. fusiformis* factor distinctly increased, while the *N. iridea- P. osloensis* factor of the EMB046/20-3GC record shows less variability (Fig. 3).

## 6 Discussion

### 6.1 Productivity changes in the last millennium

All dominant species in our benthic foraminiferal assemblages, grouped into factors, have documented association with quality (e.g. fresh or decaying) and availability of organic matter at the sea floor (e.g. Conradsen et al., 1994; Alve and Murray, 1995, 1997; Alve, 2003; Gustafsson and Nordberg, 2001; Duffield et al., 2015). The absolute abundance of planktonic foraminifera, stable carbon isotopes and total organic carbon also inform on past variability of productivity. We combine these proxies to assess productivity changes in the Skagerrak. Two periods with different productivity in the Skagerrak region are identified: i) moderate productivity between ~ CE 900 and 1700 and ii) high productivity from ~ CE 1700 towards present (Figs. 3, 5). For each defined period, we discuss the level of and changes in primary productivity and potential causes behind this productivity variability. Throughout the discussion, we also refer to the smoothed data of stable isotopes and Mg/Ca-derived BWT records as palaeothermometry proxies.

### 6.1.1 Moderate primary productivity (~ CE 900 – 1700)

The highest absolute abundance of planktonic foraminifera and a clear dominance of the *P. osloensis* factor are recorded between ~ CE 900 and 1700, indicating a period of moderate primary productivity (Fig. 3). The TOC values and $\delta^{13}$C values do not show any major changes within this interval and until ~ CE 1500, respectively (Fig. 5). High abundance of planktonic foraminifera is strongly correlated with nutrient-rich water, making them a good proxy for productivity changes (Boltovskoy and Correa, 2016). The *P. osloensis* factor includes the species *P. osloensis* and *N. iridea*, with an addition of *C. laevigata* in EMB046/10-4GC, in line with the previously identified *C. laevigata – P. osloensis* cluster (Erbs-Hansen et al., 2011) and *N. iridea – C. laevigata* category (Alve, 2010). These three species have an ecological preference for nutrient-rich environments, preferably with oxic bottom water conditions (Alve, 2003, 2010; Duffield et al., 2015). Hence, overall nutrient-rich conditions likely prevailed in the Skagerrak during this period.

Furthermore, the *B. skagerrakensis* factor and palaeoproductivity fauna also peak occasionally in the early and the late part of this interval (~ CE 900 – 1200, ~ CE 1600 – 1700) (Figs. 3, 5). The *B. skagerrakensis* factor relates to ecological preferences of the epifaunal to shallow infaunal benthic species *B. skagerrakensis*, a species associated with high fresh phytodetritus fluxes to the sea floor accompanied by a continuously high oxygen content in the sediments (Duffield et al., 2015). The abundant occurrence of *B. skagerrakensis* in the Skagerrak and Oslofjord area is restricted to water masses with temperatures between 5 and 7 °C and salinity around 35 PSU (Qvale and Nigam, 1985 and references therein; Alve and

Murray, 1995, 1997; Duffield et al., 2015). In contrast to *P. osloensis*, *N. iridea* and *C. laevigata*, taxon *B. skagerrakensis* does not feed on decaying organic matter but prefers freshly settled algal material (Duffield et al., 2015). Hence, appearance of *B. skagerrakensis* between ~ CE 900 and 1200 and ~ CE 1600 – 1700 in both records, likely indicates a period of well-oxygenated bottom water conditions with high fresh phytodetritus fluxes, while the period from ~ CE 1200 to 1600, characterised by dominance of *P. osloensis* factor (Fig. 3), suggests that bottom water oxygen conditions in the Skagerrak were somewhat less favourable for *B. skagerrakensis* or/and a change to a more food-competitive environment where the herbivorous *B. skagerrakensis* was likely outcompeted by the more omnivorous to detritivorous species *C. laevigata*, *P. osloensis* and *N. iridea*, which all are able to feed on both fresh and decaying organic matter (Alve, 2010; Duffield et al., 2015).

Because the stable carbon isotope composition recorded in calcareous benthic foraminiferal shells can be used to reconstruct past bottom water environments modified by fluxes of organic matter (Rohling and Cooke, 1999; Ravelo and Hillaire-Marcel, 2007), corresponding changes of the benthic $\delta^{13}C$ similar to those in foraminiferal assemblages would be expected. Marine organisms preferentially take up more $^{12}C$ than $^{13}C$ in their biomass. When this organic matter disintegrates after it is deposited at the ocean floor, more $^{12}C$ is released to the surrounding water. Hence, enhanced degradation at the bottom, e.g. related to enhanced primary productivity in the surface waters, will increase the $^{12}C$ concentrations in the bottom/pore water, in addition to increasing the nutrient content. Foraminifera that calcify in such a $^{12}C$ enriched water mass will record lower $\delta^{13}C$ values than if less degradation of organic matter took place (Ravelo and Hillaire-Marcel, 2007; Filipsson and Nordberg, 2010). Hence, while the changes in the benthic and planktonic foraminiferal assemblages at ~ CE 900 – 1700 suggest overall constant nutrient-rich conditions characteristic for the Skagerrak region, the $\delta^{13}C$ display light values and little variability until ~ CE 1500, supporting our interpretation of moderate primary production within the interval. This corresponds to studies by Hebbeln et al. (2006), who recorded increasing $\delta^{13}C$ values after a period of high productivity in the southern Skagerrak and relatively little variability in $\delta^{13}C$ in the northern Skagerrak between ~ CE 700 and 1500.

### 6.1.2 Causes of moderate primary productivity (~ CE 900 – 1700)

Within the period of moderate primary production common for both sites, the δw, BWT and $\delta^{18}O$ records reveal low correlation between both cores, pointing to different water conditions in the central and western Skagerrak (Fig. 5). At the time of higher BWT in the earliest ~ 250 years of the EMB046/20-3GC record followed by decreasing temperature trend until ~ CE 1450, the temperature in EMB046/10-4GC is lower and less variable between ~ CE 900 and 1350 after which it increases. The two events of most contrasting temperatures between the cores were found at ~ CE 900 – 1100 and ~ CE 1350 – 1500. In the first instance, warming in the central Skagerrak seen from higher BWT is not indicated by the $\delta^{18}O$ which instead shows higher values and corresponds to decreases in δw. Since the $\delta^{18}O$ can be induced by salinity and temperature, periods of good correlation between $\delta^{18}O$ and δw but contradictory to the BWT pattern may suggest salinity influence on the temperature signal (Brückner and Mackensen, 2006). In contrast, good correlation between all three proxies is believed to give a fair estimation of temperature and salinity changes (Fig. 4). During the second instance (~ CE 1350 – 1500), there is a general good correlation

between the BWT changes and variability of $\delta^{18}O$ values, showing similar patterns of higher BWT and $\delta w$, corresponding to drop in $\delta^{18}O$ values in EMB046/10-4CG records, and low BWT and $\delta w$ when $\delta^{18}O$ values increase in EMB046/20-3GC (Fig. 4). These changes reflect colder bottom water temperature and lower salinity from ~ CE 1350 to 1500 in the central, than in the western Skagerrak, which interestingly coincide with an event of minimum surface salinity in the northeastern Skagerrak

interpreted as enhanced outflow of low saline Baltic Sea water (Hebbeln et al., 2006). Since our cores were retrieved bellow 400 m within the deep waters of Norwegian Trench we would expect to obtain similar temperature and salinity signals for both sites. Instead, the cooling observed in central Skagerrak most likely resulted from a renewal of the deep water by inflowing colder and denser North Sea waters which apparently did not reach the shallower located EMB046/10-4GC (Ljøen and Svansson, 1972). Thus, it is possible that the higher BWT of EMB046/10-4GC rather reflected the temperature of warm

Atlantic water occupying the western Skagerrak basin (Brückner and Mackensen, 2006; Butruille et al., 2017).

The above described changes in oxygen isotopes and BWT, do not appear to correspond to variability seen in foraminiferal assemblages and the $\delta^{13}C$ records. While it is difficult to find a good match between changes in palaeoproductivity and palaeotemperature proxies, the high absolute abundance of planktonic foraminifera recorded at both sites at ~ CE 900 – 1700 (Fig. 5) suggests that the primary productivity was driven by nutrient-rich Atlantic water and abundant

phytodetritus fluxes rather than enhanced nutrients entering the area through the Baltic outflow feeding the NCC. This interpretation is supported by a study from the Northern North Sea, where Klitgaard-Kristensen and Sejrup (1996) argued that the Atlantic water is favourable for planktonic foraminifera, while the low salinity of the NCC reduces their abundance. Our interpretation is further supported by previous studies based on foraminiferal (Erbs-Hansen et al., 2011) and diatom (Gil et al., 2006) assemblages, as well as a multiproxy study by Hebbeln et al. (2006), which all report on an onset of enhanced Atlantic

water advection to the Skagerrak at ~ CE 900. Gil et al. (2006) documented an increase in diatom species associated with high salinity water in the Skagerrak and argued for enhanced inflow of nutrient-rich water via the NJC. Moreover, water with suspended sediments and low salinity are not favourable for planktonic foraminifera (Murray, 1976), therefore the higher absolute abundance of planktonic foraminifera in EMB046/10-4GC than in EMB046/20-3GC can be explained by an advantageous exposure to Atlantic water and smaller contribution of the low salinity Baltic Sea water within the NCC in the

western, than in the central Skagerrak. It has to be noticed, however, that planktonic foraminifera, due to their ability to inhabit the water column down to ~ 200 m (for species found in our records), can reflect the character of both surface and upper intermediate water layers (Jonkers et al., 2010; Schiebel et al., 2017).

Differences in water conditions between both sites are further supported by the differences in foraminiferal assemblages at CE ~ 900 – 1700 between the cores. It is likely that the higher abundance of *C. laevigata* in core EMB046/10-

4GC than in the EMB046/20-3GC reflects a higher contribution of well-oxygenated Atlantic waters to the western Skagerrak as compared to its central part (Fig. 3). *C. laevigata* and *P. osloensis* are mostly recorded in the Skagerrak and Norwegian Trench area and are associated with Atlantic water influence (Van Weering and Qvale, 1983; Conradsen et al., 1994; Alve and Murray, 1995; Klitgaard-Kristensen et al., 2002; Wollenburg et al., 2004). In EMB046/20-3GC, *C. laevigata* is largely replaced by *N. iridea* which is commonly present in the Skagerrak and the Scandinavian fjord waters with a salinity >35 PSU and a

temperature range of 6 – 6.5 °C (Polovodova Asteman et al., 2013 and references therein). In addition, *N. iridea* is capable of growth under hypoxic-suboxic conditions (Duffield et al., 2015). Hence the dominance of *N. iridea* over *C. laevigata* in EMB046/20-3GC may be related to less favourable bottom water oxygen conditions in the central Skagerrak which is more exposed to the brackish and nutrient-rich water of the BC as well as to enhanced river runoff.

5      Based on the combined high absolute abundance of planktonic foraminifera, intermediate abundance of benthic palaeoproductivity species, occasional peaks in the *B. skagerrakensis* factor, no major changes in TOC and relatively little variability in $\delta^{13}$C we conclude that the time interval CE 900 – 1700 was characterised by moderate palaeoproductivity in the Skagerrak. Furthermore, we argue that palaeoproductivity does not show coherence to changes in palaeothermometry proxies (Mg/Ca, $\delta^{18}$O) and that the moderate productivity at that time was largely sustained by nutrient-rich Atlantic water bathing the sites as deduced from the appearance of planktonic foraminifera.

      From the discussion above it is seen that several processes took place between ~ CE 900 and 1700, including changes in bottom water circulation, oxygen and salinity fluctuations or carbon fluxes. Each of these may in turn have been influenced by anthropogenic, climatic and/or oceanic factors (Brückner and Mackensen, 2008; Filipsson and Nordberg, 2010). Interestingly, the beginning of the moderate productivity period characterised by peaks in the *B. skagerrakensis* factor and palaeoproductivity species (~ CE 900 – 1200) corresponds well to the overall stable and relatively warm temperatures at the Northern Hemisphere (CE 830 – 1100) (PAGES 2k Consortium, 2013) associated with the early stage of the Medieval Climate Anomaly (MCA) (e.g. Hass, 1996). Hass (1996) argued that the MCA lasted until CE 1300 in the Skagerrak area. Based on granulometric analyses he suggested that the MCA was associated with a decreased strength of south-westerly winds as a result of a more northerly located cyclonic track causing weaker bottom currents. In contrast, the North Atlantic Oscillation (NAO) reconstructions by Trouet et al. (2009), Olsen et al. (2012), Faust et al. (2016), among others, all suggest a tendency for prevailing positive NAO conditions during the MCA and hence, south-westerlies dominating the meteorological regimes during winter (Fig. 5). During a positive NAO phase, strong south-westerlies result in warm and wet winters over the Northern Europe (Hurrell, 1995; Hurrell et al., 2001; Trouet et al., 2009) and coincide with intensification of water mass exchange (inflows and outflows) in the Skagerrak (Winther and Johannessen, 2006). Predominant positive NAO conditions would, however, also intensify the river runoff and the outflow of brackish water from the Baltic Sea to the Skagerrak due to increased precipitation over the catchment area. Increased riverine input and Baltic Sea outflow would in turn enhance the nutrient supply to the surface waters of the Skagerrak, and hence, increase the primary production. However, our palaeoproductivity proxies do not inform on increased productivity during ~ CE 900 − 1500. Neither does the high absolute abundance of planktonic foraminifera recorded at the same time in both Skagerrak sites (Figs. 3, 5) support enhanced riverine input and increased Baltic Sea outflow, since their abundance usually decreases in areas with increased brackish water conditions and decreased water transparency due to e.g. runoff (Murray, 1976; Klitgaard-Kristensen and Sejrup, 1996). Therefore, the higher winter precipitation during the MCA reported from southwestern Norway was likely blocked by the mountain ranges in southern Norway resulting in less runoff reaching our study sites (Fig. 5, Bakke et al. 2008).

The long period of high planktonic foraminiferal abundance, depleted $\delta^{18}O$ and overall warm BWT in the central Skagerrak corresponds well with a pronounced positive NAO phase reconstructed by Trouet et al. (2009) and Olsen et al. (2012) lasting until ~ CE 1450, which suggests a strong advection of warm Atlantic water (Fig. 5). Around that time, a period characterised by a deep water warming and weaker deep and cold North Sea water inflows (deep-water renewal) was suggested to take over in the Skagerrak (Butruille et al., 2017). The following drop in temperature and thus cooling in the central Skagerrak is consistent with a temperature decline in the North Atlantic around ~ CE 1400, marking the onset of the Little Ice Age (LIA) in northern Europe (Berstad et al., 2003; Brückner and Mackensen, 2006; Büntgen et al., 2011; Erbs-Hansen et al., 2011). Among other reconstructions, Trouet et al. (2009) proposed that during the LIA the NAO index became more negative, and that the associated weaker westerly airflow resulted in a reduction of the Atlantic water inflow. Reduced advection of warm Atlantic water coinciding with a deep-water renewal (discussed above) to the central Skagerrak may explain the colder bottom water conditions seen around ~ CE 1500 in EMB046/20-3GC record. The changing climate conditions during the transition between MCA and the LIA and later during the LIA, likely accompanied by increased storminess (Gil et al., 2006), were also an important additional factor behind the contradicting bottom water conditions between the central and western Skagerrak.

To conclude, it is likely that the moderate productivity during ~ CE 900 – 1450 was primarily driven by the enhanced influence of nutrient-rich Atlantic water, likely related to the predominant positive NAO associated with the warm Medieval Climate Anomaly. The Atlantic inflow was stronger than the Baltic Sea outflow, creating favourable conditions for planktonic foraminifera in the region. The second part of the moderate productivity period (~ CE 1450 – 1700) coincides with variable climate conditions, which are characteristic for the Little Ice Age, where a predominantly more negative NAO likely reduced the warm Atlantic water inflow and trigged deep-water renewal.

### 6.1.3 High primary productivity (~ CE 1700 – present)

A prominent change in benthic and planktonic foraminiferal assemblages, the $\delta^{13}C$ and TOC records took place around CE 1700 in both cores, suggesting a shift in environmental conditions in the Skagerrak. The palaeoproductivity related foraminiferal fauna increased and the values of the *B. skagerrakensis* factor largely replaced the *P. osloensis* factor (Figs. 3, 5). The planktonic foraminifera content decreases towards the top of the cores and almost disappear after ~ CE 1850 in both records. These changes in foraminiferal assemblages were accompanied by a gradual reduction of $\delta^{13}C$ and a continuous increase in TOC in both cores (Fig. 5). All proxies suggest enhanced primary productivity from CE 1700 towards present day.

### 6.1.4 Causes of high primary productivity (~ CE 1700 – present)

In consistency to the previous period, comparison of the two cores shows that the periods of high primary productivity are common for both study sites while the BWT and oxygen isotopes generally provide negative correlation in temperature and

salinity between the central and western Skagerrak. However, from ~ CE 1700 the $\delta^{18}$O and $\delta$w patterns show some similarities as comparably low values in both records and reflect increased values around ~ CE 1800, coinciding with generally low BWTs, which are again warmer at the western Skagerrak side (Figs. 4, 5). These changes suggest a colder climate followed by warmer and more saline bottom water conditions. The cooling appears during a distinctly negative NAO period showing a similar

relation between changes in the water masses and the NAO as the one seen at ~ CE 1350 – 1500. However, we suggest that at this time deep-water renewal reached to the western Skagerrak, consequently lowering temperature and salinity at both sites. The following warming of the bottom water after ~ CE 1800 reflects naturally induced environmental changes in the Skagerrak region accompanied by a gradually increasing human activity. The warming seen in our records is consistent with intensified heat transport to the Northern Hemisphere (Brückner and Mackensen, 2006), overall warm climate in Fennoscandia (Briffa et

al., 1992) and warm spring conditions recorded between CE 1750 and 1920 off the Norwegian continental margin (Berstad et al., 2003). Furthermore, our results are supported by similar $\delta^{18}$O changes recorded in the same area and at the same time by Brückner and Mackensen, (2006), and Hass (1995). The bottom water warming was associated with the termination of the coldest LIA phase in the Skagerrak region (Berstad et al., 2003; Brückner and Mackensen, 2006), a transition to a more positive NAO mode that would entail wetter and warmer winters over Scandinavia (Hurrell et al., 2001), or both. Either way, our data

demonstrate a long-term intensification of nutrient supply, likely due to increased inflows of Atlantic water to the Skagerrak. Associated with a generally more positive NAO phase it is expected that changes in the atmospheric and oceanic circulation systems would result in enhanced surface water outflow from the Baltic Sea (e.g. Gustafsson and Stigebrandt, 1996; Zorita and Laine, 2000). As a consequence of increased precipitation and thus enhanced river runoff over the large Baltic Sea catchment area, the outflowing low saline Baltic Sea water would supply the Skagerrak surface water with nutrients (Svansson,

1975; Aure et al., 1998; Krossa et al., 2015). The decreased, almost disappearing, abundance of planktonic foraminifera in our records is in line with lower salinity and transparency in the upper water layers. The total freshwater riverine discharge from the Baltic Sea together with the contributions from the major Norwegian rivers to the Skagerrak contribute with much less nutrients to the Skagerrak waters than nutrient transport via the inflows from the North Sea (Danielssen et al., 1997). However, Krossa et al. (2015) still showed a good correlation between the increased Baltic outflow and enhanced productivity in the

Skagerrak on timescales longer than 1100 years, based on increased alkenone $C_{37:4}$ concentration, a proxy for the influence of brackish water. Furthermore, Polovodova Asteman et al. (2018), documented increased palaeoproductivity over the last 1700 years that corresponded in time with the increased alkenone $C_{37:4}$ concentration (Krossa et al., 2015). Both studies argued that the nutrient-rich water causing enhanced productivity in the central Skagerrak was to a large extent of Baltic origin. Thus, the Baltic outflow can play an important role in the Skagerrak nutrient budget.

Zillén et al. (2008) showed that a widespread oxygen deficiency in the Baltic Sea was stimulated by an increased nutrient loading associated with a growing population and intensification of land use changes, which all began around CE 1600 and were followed by an industrial development at around CE 1800. In more recent times (after ~ CE 1900), land use changes in Scandinavia caused an increased terrestrial runoff through either sparsely cultivated lands after massive deforestation or due to an extensive farming in southern Sweden (Zillén et al., 2008 and references therein; Kaplan et al.,

2009). Hence it is likely that superimposed on the natural variability in volume of the outflowing Baltic water, the concentration of nutrients in the outflowing water has changed after ~ CE 1700 towards present due to a gradually increased human impact.

At the same time as increased primary production caused eutrophication in the Baltic Sea, our data show a clear decrease in the $\delta^{13}C$ values at both Skagerrak sites (Fig. 5). This distinct reduction in $\delta^{13}C$ provides evidence for a change from the
ocean-atmospheric relationship established for the preceding periods towards an additional contribution of the lighter carbon isotope ($^{12}C$) to the sea water from the atmosphere due to the increase in atmospheric $CO_2$ concentration caused by anthropogenic emissions, known as the Suess effect (e.g. Cage and Austin, 2010; Filipsson and Nordberg, 2010; Eide et al. 2017). Both the increased primary productivity and the Suess effect may cause a reduction of benthic $\delta^{13}C$. However, the decrease of ca 0.9‰ seen in our $\delta^{13}C$ records from ~ CE 1800 towards present is likely to a large extent explained by the Suess
effect, which is in line with the ca 0.8‰ $\delta^{13}C$ decrease between the preindustrial and modern period observed in the North Atlantic Ocean (Eide et al., 2017).

To summarize, from ~ CE 1700 towards present day, changes in the palaeoproductivity proxies indicate increased primary productivity likely caused by an enhanced nutrient content driven by a combination of influence of nutrient rich, warm Atlantic water, enhanced Baltic outflow, intensified river runoff and enhanced human impact through agriculture expansion
and industrial development. This increase in primary production occurred during a high variability in temperature and NAO index.

### 6.1.5 Changes in the last ~ 70 years

During the last 70 years *B. skagerrakensis* started to decrease in favour of higher *S. fusiformis* abundances (Fig. 3). *S. fusiformis* is considered as an indicator of trophic changes in Scandinavian waters due to its high tolerance of oxygen-depleted and
organic-rich conditions (e.g. Alve, 2003 and references therein). This opportunistic species has the highest reproduction (up to 7 times/month) and growth rates after the phytoplankton blooms, followed by an enhanced food supply to the sea floor and decay of organic matter (Gustafsson and Nordberg, 2001). This explains the taxon's food preferences recognised as both fresh phytodetritus and microbes associated with degradation of organic matter (Duffield et al., 2015). Therefore, the increased abundance of *S. fusiformis* during the last 70 years and simultaneous drop in *B. skagerrakensis* suggest changes in the quality
of organic matter at the sea floor after high productivity episodes causing increased enrichment of organic matter and enhanced degradation in the sediments. At the same time, a continuously high nutrients content coinciding with a gradual decline in oxygen concentration has been shown for the Skagerrak fjords with sluggish bottom water circulation (e.g. Rosenberg, 1990; Johannessen and Dahl, 1996, Alve, 2003; Filipsson and Nordberg, 2004). There are no hydrographic studies reporting on low oxygen conditions in the deep Skagerrak basin during the last 70 years, thus high appearance of *S. fusiformis* in our records
will indicate rather higher amount of degraded food than depleted oxygen.

### 7 Conclusions

This study provides evidence for changes in primary productivity in the Skagerrak during the last millennium. Our multi-proxy records show that the time interval ~ CE 900 – 1700 was characterised by moderate primary production with nutrients largely sustained by warm Atlantic water, as revealed by high abundance of planktonic foraminifera. The first part of this interval (~ CE 900 – 1450) was likely associated with the warm Medieval Climate Anomaly during which a persistent positive NAO strengthened the westerlies, resulting in more frequent warm Atlantic water inflows. After ~ CE 1450, continuously moderate productivity at both sites are indicated by a dominance of the *P. osloensis* factor, high abundance of planktonic foraminifera, relatively stable TOC, and overall stable $\delta^{13}C$. This continuously moderate productivity at both sites coincided with the variable climate conditions characteristic for the Little Ice Age. Episodes of negative NAO triggered deep-water renewal resulted in colder bottom water temperature in the central Skagerrak, while the western Skagerrak seems to be more resistant to this cooling and instead reflects the temperature of warm Atlantic water that occupied this site.

Finally, the high primary productivity period between ~ CE 1700 and 2000 is documented by both increased TOC and *B. skagerrakensis* factor, high absolute abundance of palaeoproductivity fauna and decreased $\delta^{13}C$ values. Enhanced nutrient availability was likely caused by a stronger Baltic Sea outflow, increased river runoff, intensified inflows of the nutrient-rich Atlantic water, together with agricultural and industrial expansion. Simultaneously, an increase in human induced $CO_2$ emission caused great change in the oceanic carbon isotope budget, indicated by the Suess effect, shown in our records by strongly negative $\delta^{13}C$ values since ~ CE 1800. The most pronounced increase in primary production at ~ CE 1800 – 2000 occurred during a warm period with a more positive NAO, wetter and warmer winters in Scandinavia as shown by an increase in BWT and decrease in $\delta^{18}O$ in our records.

A comparison between the two records show an overall slight difference in species composition. This difference in species composition is likely due to a more favourable habitat in the western Skagerrak, with less exposure to low-saline nutrient-rich NCC water, while the higher TOC in the central Skagerrak mostly results from an exposure to the Baltic outflow, nutrient rich water reaching the site via NJC, and terrestrial runoff due to a more inland location. The productivity and the fluxes of organic matter to the seafloor appear to not correspond to the temperature and salinity changes recorded in the benthic *Melonis barleeanus* shells.

## Data availability

The presented data are available at www.pangaea.de, doi: 10.1594/PANGAEA.894153.

## Competing interests

The authors declare that they have no conflict of interest

## Acknowledgements

This research is a part of the ClimLink project, which was funded by Norway Grants: POL-NOR/199763/92/2014 in the Polish-Norwegian Programme operated by the National Centre of Research and Development of Poland. We thank the captains, chief scientists and crews of *RV Elisabeth Mann-Borgese* for logistical and technical assistance. We also thank Małgorzata Bąk (University of Szczecin) for coordinating the project. Joanna Sławińska, Ryszard Borówka and staff of the Laboratory of the Department of Geology and Palaeogeography (University of Szczecin, Poland) performed geochemistry analyses, while Are Olsen (University of Bergen) and Jeroen Groeneveld (University of Bremen) contributed with valuable comments regarding the $\delta^{13}C$ - the Suess Effect relationship and the Mn/Ca ratio, respectively. Rocio Castano Primo helped with archiving data at Pangaea. Finally, we thank Marit-Solveig Seidenkrantz and one anonymous reviewer for constructive feedbacks.

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

**Tables**

**Table 1: Information about the new chronology of core EMB046-20-3GC over the last ca. 1600 years**.

| Identification | Core | Sample depth (cm) | Based on/ Dated material | $^{14}$C date | ΔR | Calibrated age range ±1σ BP 1950 | Rel. prob | Calendar age BP 1950 (med. prob.) | Cal a. CE Tie points used | References for individual dates |
|---|---|---|---|---|---|---|---|---|---|---|
| | EMB046-20-3GC | 0 | Year of coring | | | | | -63 | 2013 | Polovodova Asteman et al., (2018) |
| | EMB046-20-3GC | 25 | Hg (μg/kg) | | | | | 50 | 1900 | Polovodova Asteman et al., (2018) |
| Poz-59813 | EMB046-20-3GC | 5.5 | Mixed foraminifera | 560±40 | 200±50 | Post bomb | | | | Polovodova Asteman et al., (2018) |
| ETH-88814 | EMB046-20-3GC | 30.5 | Mixed foraminifera | 755±50 | 200±50 | 101-273 | 1 | 178 | 1841 | This study |
| Poz-68082 | EMB046-20-3GC | 61 | Mixed foraminifera | 840±40 | 200±50 | 232-398 | 1 | 287 | 1718 | Polovodova Asteman et al., (2018) |
| ETH-87337 | EMB046-20-3GC | 85 | Mixed foraminifera | 1045±50 | 200±50 | 412-531 | 1 | 468 | 1538 | This study |
| Poz-99621 | EMB046-20-3GC | 112 | Mixed foraminifera | 1220±30 | 200±50 | 549-641 | 1 | 596 | 1401 | This study |
| Poz-68083 | EMB046-20-3GC | 141 | Mixed foraminifera | 1440±60 | 200±50 | 712-872 | 1 | 789 | 1238 | Polovodova Asteman et al., (2018) |
| Poz-99622 | EMB046-20-3GC | 200 | Mixed foraminifera | 1835±30 | 200±50 | 1144-1264 | 1 | 1196 | 806 | This study |
| Poz-99623 | EMB046-20-3GC | 274,5 | Mixed foraminifera | 2205±30 | 200±50 | 1504-1655 | 1 | 1570 | 295 | This study |

**Table 2: List of dominant (bold) and accessory benthic foraminiferal species. The species names marked by "*" represent foraminiferal species with relative abundance >5% in only one sample, thus were excluded from statistic classification.**

| Core EMB046/10-3GC |
|---|

*Brizalina skagerrakensis* Qvale & Nigam, 1985
*Bulimina marginata* d'Orbigny, 1826
*Cassidulina laevigata* d'Orbigny, 1826
*Cassidulina neoteretis* Seidenkrantz, 1995
*Eggereloides medius* (Höglund, 1947)
*Epistominella sp. including E. exigua* (Brady, 1884) *E. vitrea* Parker, 1953
*Hyalinea balthica* (Schröter in Gmelin, 1791)
*Melonis barleeanus* (Williamson, 1858)
*Nonionella iridea* Heron-Allen & Earland, 1932
*Pullenia osloensis* Feyling-Hanssen, 1954
*Stainforthia fusiformis* (Williamson, 1848)

| Core EMB046/20-4GC |
|---|

*\*Bolivina spathulata* (Williamson, 1858)
*Brizalina skagerrakensis* Qvale & Nigam, 1985
*\*Bulimina elegantissima* d'Orbigny, 1839
*\*Bulimina marginata* d'Orbigny, 1826
*Cassidulina laevigata* d'Orbigny, 1826
*Cassidulina neoteretis* Seidenkrantz, 1995
*\*Cassidulina norcrossi* Cushman, 1933
*Eggereloides medius* (Höglund, 1947)
*Epistominella sp.* including *E. exigua* (Brady, 1884) *E. vitrea* Parker, 1953
*Hyalinea balthica* (Schröter in Gmelin, 1791)
*Melonis barleeanus* (Williamson, 1858)
*Nonionella iridea* Heron-Allen & Earland, 1932
*Pullenia osloensis* Feyling-Hanssen, 1954
*Recurvoides laevigata* Höglund, 1947
*Stainforthia fusiformis* (Williamson, 1848)
*\*Triloculina tricarinata* d'Orbigny, 1826
*\*Trochammina sp.* Parker & Jones, 1859

**Table 3: The factor results from a CAB-FAC factor analyses.**

| EMB046/10-4GC | | |
|---|---|---|
| Factors | Eigenvalue | Variance (%) |
| 1 | 88.665 | 86.08 |
| 2 | 9.1984 | 8.93 |
| 3 | 1.8953 | 1.84 |

| EMB046/20-3GC | | |
|---|---|---|
| Factors | Eigenvalue | Variance (%) |
| 1 | 45.2 | 80.72 |
| 2 | 6.4351 | 11.49 |
| 3 | 1.3995 | 2.5 |

**Table 4: The varimax scores for factors 1 – 3. The bold numbers indicate foraminiferal species with absolute value of factor scores >1.**

| EMB046/10-4GC | | | |
|---|---|---|---|
| Foram. species | Factor 1 | Factor 2 | Factor 3 |
| *B. skagerrakensis* | -0.95083 | **3.1052** | 0.11006 |
| *B. marginata* | 0.28721 | 0.29398 | -0.18906 |
| *C. laevigata* | **1.3438** | 0.45052 | **-1.234** |
| *C. neoteretis* | 0.036824 | 0.15451 | -0.01801 |
| *E. medius* | 0.13292 | 0.57788 | 0.48831 |
| *Epistominella sp.* | 0.18666 | 0.15052 | 0.2312 |
| *H. baltica* | 0.18637 | 0.35011 | -0.25495 |
| *M. barleeanus* | 0.25854 | 0.14265 | -0.13135 |
| *N. iridea* | **1.0553** | 0.25077 | **1.8493** |
| *P. osloensis* | **2.4565** | 0.68666 | -0.91334 |
| *S. fusiformis* | 0.95088 | 0.10359 | **2.191** |

| EMB046/20-3GC | | | |
|---|---|---|---|
| Foram. species | Factor 1 | Factor 2 | Factor 3 |
| *B. skagerrakensis* | -0.81099 | **3.1907** | -0.01581 |
| *C. laevigata* | 0.55287 | 0.27574 | -0.19865 |
| *C. neoteretis* | 0.030673 | 0.23185 | -0.11031 |
| *E. medius* | 0.38736 | 0.13685 | -0.15638 |
| *Epistomienlla sp.* | 0.26935 | 0.1446 | 0.47431 |
| *H. baltica* | 0.31697 | 0.33722 | 0.73552 |
| *M. barleeanus* | 0.30047 | 0.062139 | -0.09968 |
| *N. iridea* | **1.8606** | 0.35752 | **2.357** |
| *P. osloensis* | **2.3155** | 0.54091 | **-2.0843** |
| *R. laevigatum* | 0.202 | 0.007055 | -0.31694 |
| *S. fusiformis* | 0.87063 | 0.33472 | 0.38467 |

**Figures**

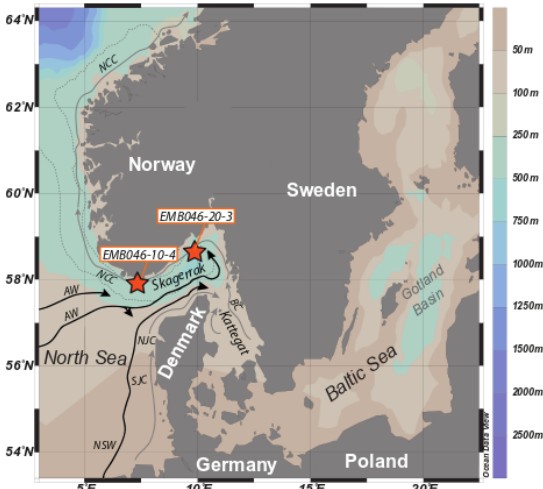

**Figure 1: Map of core locations (stars) in the Skagerrak, the NE North Sea (modified from Polovodova Asteman et al. 2018). The Norwegian Trench is outlined by a thin grey dotted line along the coast of Norway. Major current systems and water masses in the Skagerrak are indicated by arrows: the Baltic Current (BC), Northern Jutland Current (NJC), Southern Jutland Current (SJC), Norwegian Coastal Current (NCC), North Sea Water (NSW) and Atlantic Water (AW). The light grey arrows show surface water and the black arrows show subsurface water.**

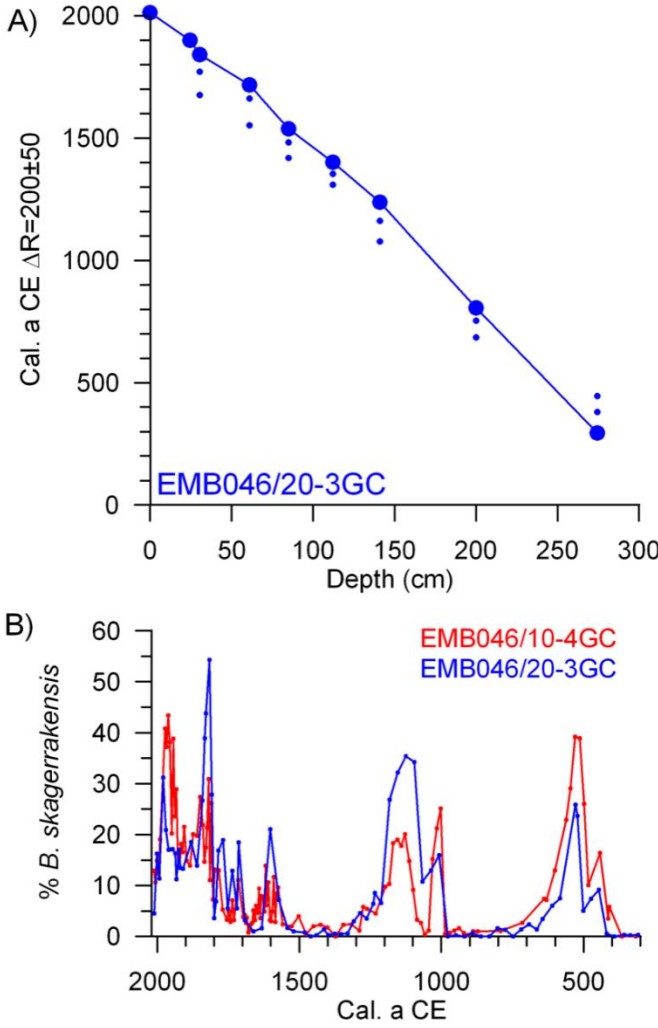

**Figure 2: (A)** Linear interpolation between the established tie points for the EMB046/20-3GC age model. **(B)** The relationship between *B. skagerrakensis* of EMB046/10-4GC (red curve) and EMB046/20-3GC (blue curve).

Fig. 3

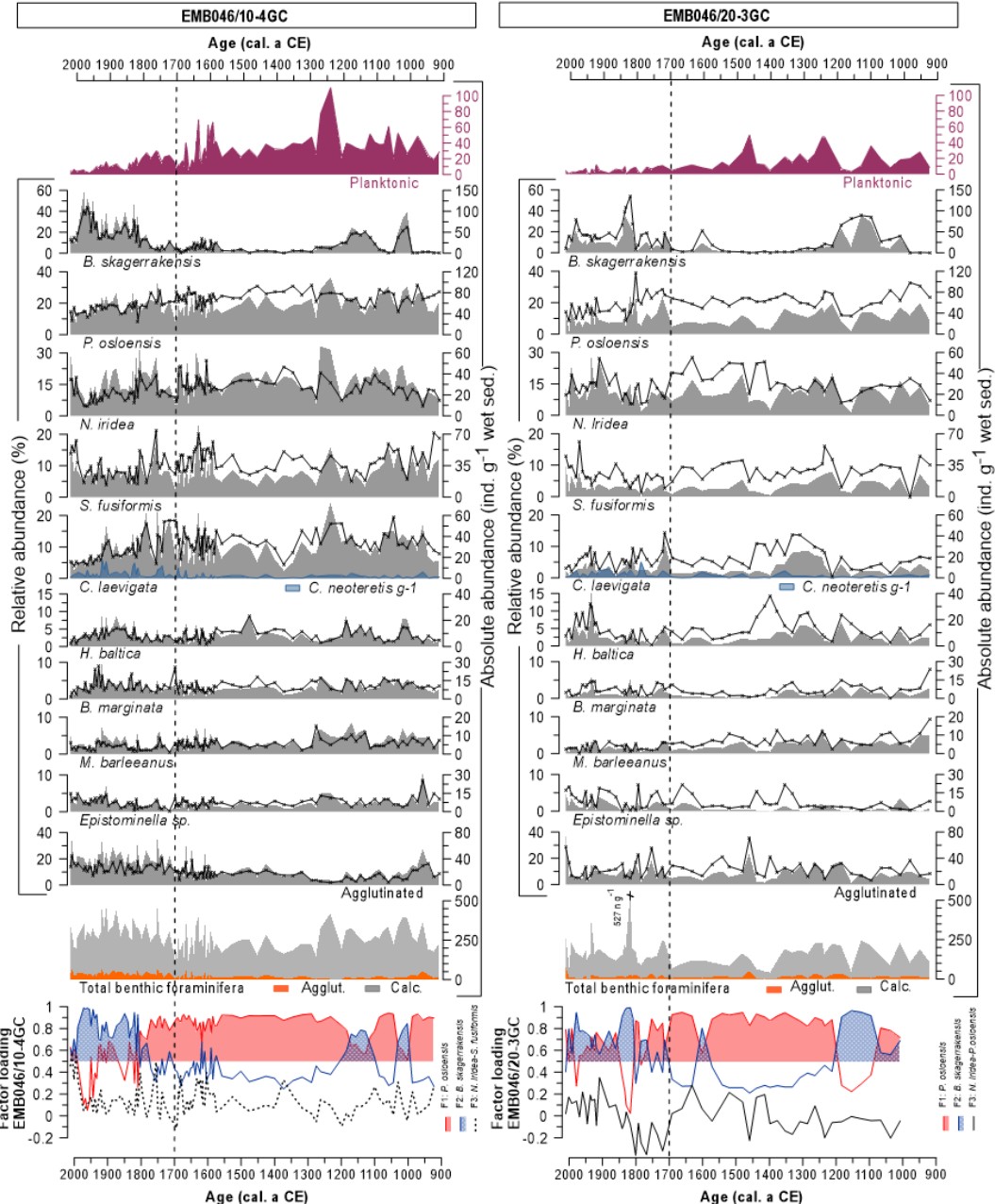

**Figure 3: Foraminiferal assemblages including dominant and accessory benthic species for both cores (EMB046/10-4GC and EMB046/20-3GC) and CABFAC results. Absolute abundance is shown as grey filed, while relative abundance as black curve with symbols. The absolute abundance of total benthic foraminifera is a sum of all species: agglutinated (Agglut.) and calcareous (Calc.). Dashed line divides record into 2 periods of most pronounced palaeoproductivity changes, which are discussed in the text.**

# Fig. 4

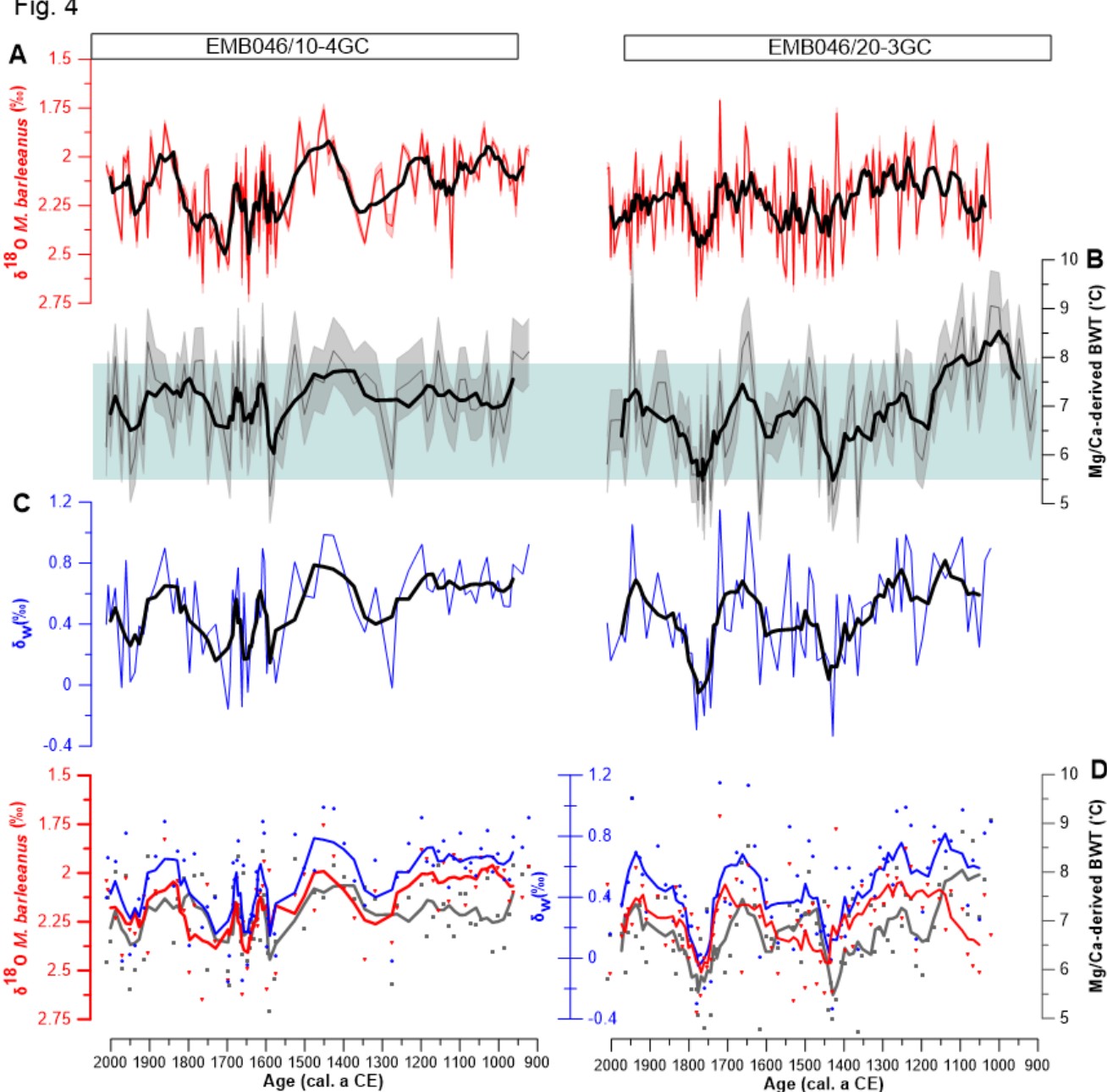

**Figure 4: (A)** Stable oxygen isotopes (δ[18]O); **(B)** Mg/Ca-derived bottom water temperature (Mg/Ca-derived BWT); and **(C)** stable oxygen isotope composition of sea water (δw) from both studied cores against age. The errors bands represent 1 SD uncertainties of the records. **(D)** The same parameters as in A-C (δ[18]O, Mg/Ca-derived BWT, δw), however, only measurements from core depths where all proxies are available are shown (symbols). The curves correspond to 5-point running average. The light blue box indicates range of instrumentally recorded temperatures of the time period between 2009 and 1924 years from the area between 57°.17–58°´N and 8°–9°.79´E at 300 – 340 dbar (ICES 2010).

Fig.5

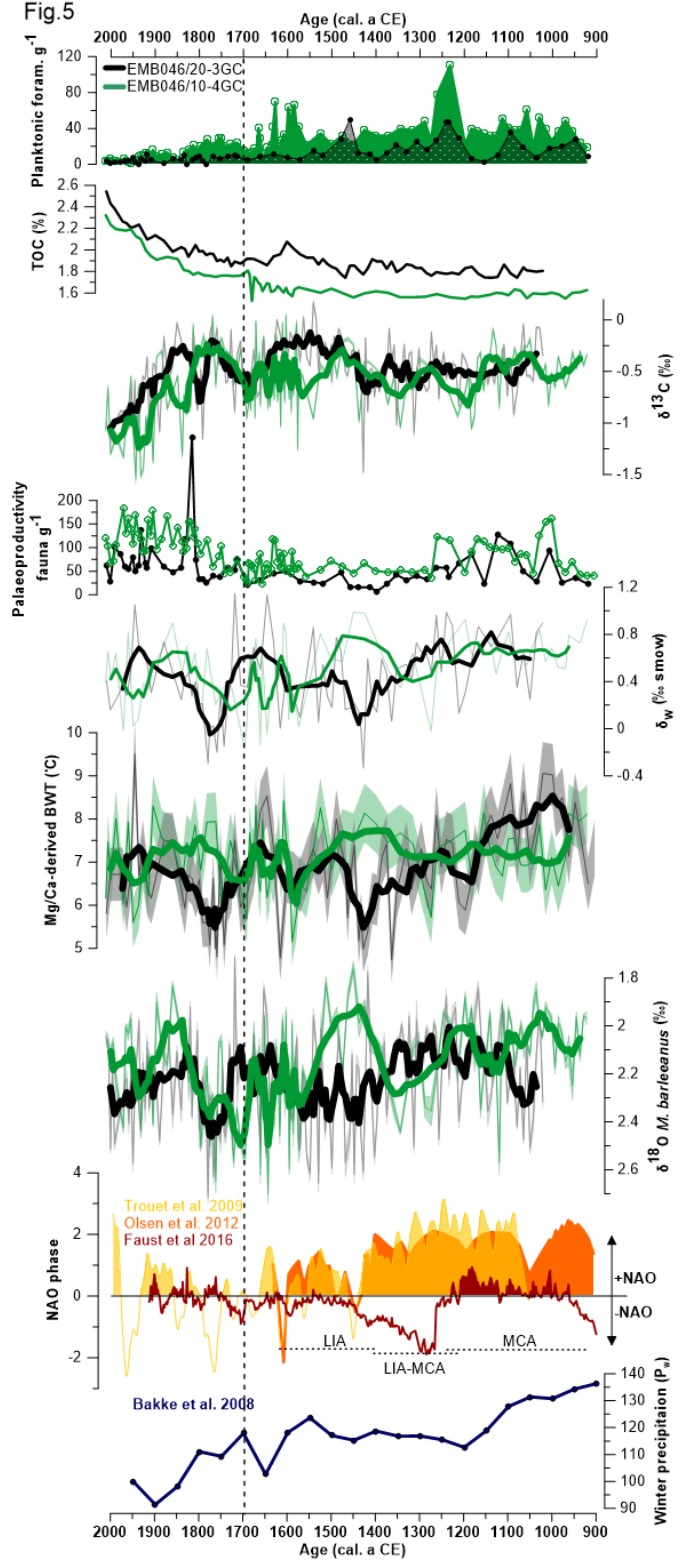

**Figure 5: Comparison of absolute abundances of planktonic foraminifera, total organic carbon (TOC), stable carbon isotope ($\delta^{13}$C), absolute abundance of palaeoproductivity fauna, the CABFAC results, stable oxygen isotope composition of sea water ($\delta$w), Mg/Ca-derived bottom water temperature (Mg/Ca-derived BWT), stable oxygen isotopes ($\delta^{18}$O) of the two studied sediment cores EMB046/10-4GC (green curves) and EMB046/20-3GC (black curves).**
5   **Reconstructions of the NAO index (yellow curve – Trouet et al (2009), orange curve – Olsen et al. (2012), red curve – Faust et al. (2016)) and winter precipitation (blue curve – Bakke et al. (2008)). The thicker curves correspond to 5-point running average. The errors bands represent uncertainties of the records. Dashed line divide record into 2 periods of most pronounced palaeoproductivity changes, which are discussed in the text.**