# Peer review of "Coastal primary productivity changes over the last millennium: a case study from the Skagerrak (North Sea)."

_Biogeosciences, 2018_

## Referee Comment (RC1) · Anonymous Referee #1 · 27 Mar 2018

General comments

The paper presents a new data set from two high-resolution cores in the Skagerrak, spanning the last 1.1 thousand years, with the purpose of investigating historical changes in productivity. The data itself is a nice contribution to our understanding of recent changes in the ecosystem and forms a good follow-up to previous work from the same group and is very relevant to the topics covered by Biogeosciences. However, I have some concerns about the overall discussion and interpretation of the data. Additional data is not needed, but the existing data should be more carefully reconsidered. In particular, the main conclusions about changes in productivity appear to rely closely

on only one part of the data set i.e. the benthic foraminifera records. The authors identify three periods of different productivity based on some good analyses of the changes in benthic assemblages, but then appear to try to match each of the other records (TOC, $\delta$13C, Mg/Ca, $\delta$18O, Mn/Ca) to these three periods, which in many cases does not seem appropriate based on the figures provided. For example, in description of the Mg/Ca results, the authors highlight high temperatures between 900 and 1200 AD, I suppose because 1200 AD is where they have identified a benthic change. However, in my opinion, for the core EMB046/10-4GC this period does not stand out, and for the core EMB046/20-3GC the temperatures seen in the earlier period continue until perhaps 1400. More careful analysis and description of this and the other records should be provided. In addition, some of the speculation regarding sources of water/nutrients to the area (particularly the Atlantic Water) needs some more thought, and could be improved by trying some additional calculations based on the existing data. Uncertainties in the data need to be better characterised in the text and figures. Also, I think the authors could use the introduction to better highlight the importance of their work for a reader not familiar with the Skagerrak.

Specific Substantial Comments

Motivation

As a reader unfamiliar with the Skagerrak, I felt that more was needed in the introduction to explain why the work is important and to highlight the interest of the work to a wider audience. There are references to changes in the ecosystem, but it would help the reader a lot to know more specifically why changes in production are important to the region (e.g. fisheries etc). Perhaps put this motivation up front before the more technical introduction.

Uncertainty

The uncertainties on each record need to be better characterised throughout the paper. In particular, uncertainties should be quoted for all methodologies, and the level of

confidence indicated. At present it is unclear whether the errors that are quoted are at 1 or 2 sigma, and this makes a huge difference. It would be helpful to have a bar showing the uncertainty beside each record in the figures as well.

Pg 5 Line 20: What is the TOC uncertainty? Even though it is published elsewhere it wouldn't take much space to include it here. Just a sentence to say uncertainty is 0.01 % (confidence).

Pg 6 Line 3: What is the level of confidence for these uncertainties? And are they long term precision or just the precision for the runs?

Pg 6 Lines 19-27: Discussion of errors needs to be clearer in this section. Line 19: what is the 0.016 long term precision measured on (I assume a standard material) and what is it's Mg/Ca value? What is the confidence level? Line 20: Join these two sentences together (e.g. 'show a good reproducibility, with the pooled. . .'. Line 22: This is ok but the authors still need the confidence level. Lines 25-27: I would like to see an estimate of the typical temperature uncertainty obtained when propagating Mg/Ca ratios through the calibration equation here. According to the Hasenfratz paper, the $2\sigma$ uncertainty ranges from $\pm$ 0.6 to 2 oC depending on the temperature. I guess we might be looking at $\pm$ 1.5 oC, and most of the wiggles in Figure 3 might fit within these uncertainty bounds. The authors really need to make sure that the moving average record is real and not just random scatter.

Pg 6 Line 28: I am less concerned that errors on counts would be a big problem for the results, especially for those with the biggest signals. However, it would still be nice to have an estimate of the counting uncertainty for each species. Ie. Replicate a sample several times how close can you repeat the counts? There are also some more formal ways to estimate counting uncertainty. See e.g. (Heslop et al. 2011, Diagnosing the uncertainty of taxa relative abundances derived from count data, Marine Micropaleontology 79, pp 114-120).

Pg 7 Line 10: It would be useful to know what the typical age uncertainty for a given

depth might be for both cores. The period after 1900 looks well-constrained by Hg, but radiocarbon can be difficult to use to match two cores exactly. However, I don't think the age model should affect your results too much. My main concern would be in saying things like 'between 1550 and 1650 one core shows this while the other core shows that.' Can the cores really be associated to that level of accuracy?

Describing the records in terms of the three periods

This was my main area of concern when reading through the paper. Firstly, I note that the three time periods loosely correspond to different sampling resolutions (i.e. relatively high resolution for the early and late periods and relatively low resolution for the middle period). Could this be affecting where lines were drawn? I don't think it is a big problem but it could be worth some investigation or a note in the text.

Secondly, as I noted above, the most convincing case for separating the record into three periods seems to be the faunal data. If I were to ignore those data and to try to describe the temperature, TOC, $\delta13C$ and Mn/Ca I would not find the patterns described in the text. The description of the temperature signal does not seem very objective. I gave one example above, and I note again how the record for EMB046/10-4GC doesn't seem to have any period that is very different, especially once you include the temperature uncertainty. I can see a period of low temperature for the other core between maybe 1450 and 1650-1700, but I do not see why we would distinguish the first 200 years from the next 500. The temperature and oxygen isotope records need a more thorough and objective analysis of whether they do actually match the faunal data, especially as the authors use such a match to argue for changes in water mass driving the faunal changes.

On Pg 12, Line 20 the authors describe the TOC as low, during the period of low productivity. However, it is higher than in the first period (argued to be moderately productive) and where the authors have also argued for lower oxygen concentration based on Mn/Ca. I find it hard to see how you could have less production, more bottom water

oxygen and yet higher TOC if TOC is being considered as a productivity proxy. How about the possibility that the sediment TOC gets less down the core simply because it is decaying over time?

Influence of different water masses

Given the issues I have outlined above, the inferences that different water masses are influencing the sites at different time (eg. Atlantic Water contributing to moderate productivity during the early period), are not well-supported. For example, the authors argue that warmer conditions (from Mg/Ca and $\delta$18O) during the 900-1200 period can be explained by increased Atlantic Water. Firstly, I suggest that the authors need to robustly test their trends with appropriate errors as outlined above. Secondly, the arguments based on temperature (especially in this section, Pg 11) are split by a section on carbon isotopes, which confuses the flow of the paper. There seem to be two arguments for temperature change: increased Atlantic Water influence and generally high temperature during the MWP. I think the authors do combine these arguments by reference to NAO changes, but the link is not made very clear. In addition, the authors should at least address why the temperature in core EMB046/20-3GC remains high well into their second time period, when the productivity has already decreased, which they overall argue is a partly a response to less AW. There is some discussion of complicating factors that might affect the temperature/productivity link, but it is not very clear and should be restructured. To solidify the link to increased AW I would suggest using the Mg/Ca and $\delta$18O to reconstruct the $\delta$18O of the deep water (e.g. salinity), because the $\delta$18O of the shells also has a temperature signal. Knowing the water $\delta$18O might provide a more useful indicator of water provenance and is standard practice in most paleoceanographic work, although the uncertainty might again be very high in this case. The references used to support the conclusions here seem appropriate, and they may be the simplest and best way to argue that the changes seen are do in part to water masses.

$\delta$13C records

The authors attempt to put the $\delta13C$ records into the context of their productivity results, and they note correctly that the late increase in $\delta13C$ is due to the Suess effect. However, given that changes in water masses are invoked to explain changes in productivity, I would expect to see some analysis regarding whether their $\delta13C$ changes could be explained this way, rather than by changing productivity at the sites. I think the authors correctly conclude that the $\delta13C$ records do not correspond to the productivity records. Therefore, I would recommend removing much of the $\delta13C$ discussion, simplifying it to a sentence or two saying that not much coherence is seen. I also think the Suess effect is well known enough that such a long paragraph from Pg 16 line 10 is not really warranted. Cutting out the $\delta13C$ discussion would also help streamline the paper.

Specific Minor Comments

Pg 3 Line 6: Note that in general, not all organic matter makes it to the seafloor (in fact most doesn't), and much of it is remineralised in the water column. If this situation is different in the Skagerrak the authors should say so and why.

Pg 3 Line 9: Sentence reads a little strangely. Consider rephrasing to something like 'Via photosynthesis, primary producers can help to remove CO2 from the atmosphere as part of the biological pump'.

Pg 3 Line 11: Change to 'likely negatively impact'.

Pg 3 Line 13: Sentence needs rephrasing. Need to specify 'export of organic matter' rather than 'production'.

Pg 3 Line 18: Include Figure 1 reference here to guide readers who don't know anything about the Skagerrak.

Pg 3 Line 20: How does increased air-sea gas exchange lead to increased nutrients? Perhaps expand a little.

Pg 3 Line 23: 'positive impacts on growth'. Growth rate? Growth magnitude? Be more

careful about saying 'positive impacts'. Do you mean positive in the sense of being 'good', or simply 'more'?

Pg 3 Line 24: Similar to above, what is a negative change in trophic levels? Fewer trophic levels? This needs to be more specific.

Pg 4 Line 7: At the first introduction of the Kattegat, provide reference to Figure 1.

Pg 4 Line 9: It might be better to say that 'in the past the NAO has represented one of the leading modes of natural climate variability over the North Atlantic'. Or remove the word 'natural' altogether. I think it may now be difficult to say whether the NAO is wholly 'natural' anymore.

Pg 4 Line 12: Specify what you mean by BCE.

Pg 4 Line 26: 'governed by' could be more specific. How does the sill control the AW?

Pg 5 Line 2: 'internal' might not be the best word here. Internal to what? Perhaps say 'In addition to the mean circulation, processes such as. . .' Also as a general point, it helps the reader if all the references can be at the ends of sentences where possible. It is possible in this instance.

Pg 5 Line 16: There are two sentences here that are almost the same. Consider rephrasing to avoid repetition.

Pg 5 Line 25: Avoid the use of 'proved' (or proven). I think it is fine to just say 'potential'.

Pg 6 Line 1: What grade of methanol was used?

Pg 6 Line 1: 'For each measurement 2-4 specimens were used.' How many measurements were done for each depth? Are 2-4 specimens enough to get a robust average isotope signal?

Pg 6 Line 10: I think the authors could be more specific about the grade of HNO3 used. What Ca concentration were the final solutions? Were these matched to the standard

concentrations?

Pg 6 Line 14: These results do not indicate 'no contamination', only 'no systematic contamination'. The authors are correct in the following sentence where they indicate that because of their high Mn/Ca values, contamination may be an issue. No need for the sentence on Line 16, as it repeats what was already said.

Pg 6 Line 18: 'After every 8 samples'? 'For' might not be specific enough.

Pg 6 Line 18: Which standard solutions were used?

Pg 6 Line 28: 'The foraminiferal faunal analysis'

Pg 7 Line 14: Include 'IntCal' in the wording somewhere here.

Pg 7 Line 21: What is the detection limit for the TOC measurements? I think the results look fine but might be worth checking.

Pg 7 Line 26: Here is an instance where knowing what the relative age uncertainty when comparing the different cores could be important. 1550-1650 is not a very long time period in terms of typical radiocarbon uncertainties.

Pg 7 Line 28: To me 'long-term trends' implies a gradual change to different conditions over the whole record. Instead these records are quite flat until the Suess effect kicks in. I might say 'long-term variations'.

Pg 8 Line 6: Same comment as above but now for oxygen isotopes.

Pg 8 Line 26: I know that identifying planktonic forams is very difficult at the tiny sizes looked at here. I agree with the author's decision to discuss only the total number rather than species. Do the benthic ID's suffer from the same difficulty. I have less benthic experience to offer a full comment here.

Pg 9 Line 7: What is meant by 'less fluctuating'? Apart from one very large peak in 20-3GC the records look quite similar. Are the standard deviations very different?

[Figure]

Pg 9 Line 9: How does changes in mass accumulation affect the results? When looking at forams per gram, two things can change the magnitude: 1) number of forams and 2) grams of sediment. Can the authors rule out changes in mass accumulation rate with their age model?

Pg 9 Line 18: This statement seems quite broad. Several of the low abundance benthics do not seem to show any trends (perhaps because of the counting uncertainty?). Some of them show different trends. I think B. skagerrakensis can be highlight more here, because it is really the main record that shows any large and consistent changes.

Pg 10 Line 6: This first sentence needs rephrasing. What is meant by 'quality'? Again avoid 'proven'. Is it the individual species or the 'factors' that have an association with organic matter? I don't think the wording makes it clear.

Pg 10 Line 15: The following 3 sections should be rewritten taking into account the substantial comments above, taking more care over the uncertainties in each record, and appreciating more that e.g. temperature and oxygen isotopes don't necessarily match the benthic productivity records.

Abstract and conclusion: Can be looked at again after more careful consideration of the data.

Figure 2: n g-1 I assume to mean number per gram. It could be misunderstood as 'per nanogram' so perhaps change to 'Number g-1'.

Figure 4: For the uninitiated like me, 'Factor loading' doesn't mean very much as an axis title. Perhaps explain a bit more in the text.

Technical Corrections

There are a large number of instances where the language could be tightened up in the paper that would help make it easier to read. I would suggest having the paper thoroughly proof read again. Previous papers from some of the same authors read well. I have put a very few points below.

Pg 5 Line 29: Change 'since' to 'because'. I have had it pointed out to me before that 'since' can be a confusing word in a paper dealing with the past.

Pg 6 Line 12: Rephrase sentence to 'Fe/Ca and Mn/Ca ratios show no correlation with measured Mg/Ca values.'

Pg 6 Line 26: Hasenfratz miss-spelled.

Pg 9 Line 8: Change 'sticking out' to something sounding more scientific.

Figure 3 caption: '5-point' not '5-points'

---

## Referee Comment (RC2) · Anonymous Referee #2 · 31 Mar 2018

The paper presents two records of benthic foraminiferal assemblages and geochemical analyses from the northern deep Skagerrak region. The records are excellent, presenting high-resolution data for the last millennium. The strength of the paper is especially the fact that data from two neighboring cores are presented, as provides evidence of a general pattern. The paper is overall well-written and clearly presented and I believe that it fits well within the scope of the journal.

However, the ms has a tendency to focus too much on local conditions and comparison to relatively few previous studies. It would therefore benefit from including information from a broader range of study sites as well as from other types of records, including

terrestrial and lacustrine records. Also more direct comparisons between records as well as an improved precision of the discussion of water masses is needed. Finally, not all data (e.g. Mg/Ca) are actually used to any significant extend in the discussions:

A key element of the paper is the link between the record and the North Atlantic Oscillation (NAO). A number of studies have suggested a more positive phase of the NAO during the MCA and a negative phase during the LIA. However, the here the authors only refer to one study without taking into account that other, earlier, studies have also made this suggestion (e.g. from off Portugal, in the Labrador sea/West+East Greenland etc.). Also, since the present manuscript provides a high-resolution records, these data should in fact be plotted vs. the high-resolution NAO reconstructions (Trouet et al, as well as vs. other high-resolution records such as those by Olsen et al 2012 and Faust et al 2016). It would be very interesting to see, if the overall quite well-known trend of positive NAO during the MCA and negative NAO during the LIA is also seen at shorter, decadal/multidecadal time scales. Whether such a correlation between productivity and NAO cannot be verified, it would be valuable information. Olsen, J., Anderson, N.J, Knudsen, M.F, 2012: Variability of the North Atlantic Oscillation over the past 5,200 years. Nature Geoscience 5, 808–812, doi:10.1038/ngeo1589 Faust, J.C., Fabian, K., Milzer, G., Giraudeau, J., Knies, J., 2016: Norwegian fjord sediments reveal NAO related winter temperature and precipitation changes of the past 2800 years. Earth and Planetary Science Letters 435, 84-93.

The study region is influence by several different water masses, including local outflow of low-salinity water from riverine outflow, saline Atlantic water and more intermediate salinity water as a mixture of North Sea and riverine waters (e.g. the Jutland Current). However, in the presentation of the water masses, it is not always clear at which levels in the water column these water masses are found, nor whether they also influence the actual study sites. This problem continues throughout the discussion and the one gets the impression that either there is increased Atlantic water or increased low-salinity water. However, stratification could allow both. Thus, the discussion need to be much

more precise. In this context, I do agree that the increased flux of planktic foraminifera indicated increased inflow of Atlantic water. However, how sure are the authors that the planktic foraminifera are in fact locally produced and not brought in from the Atlantic via the currents? The planktic foraminifera may not be autochthonous and even if they are, they would likely not represent direct surface waters. Thus, this inflow may not have occurred right at the surface, but rather as a subsurface current, thus still allowing an increased surface-outlow of lower-salinity waters. In the discussions on whether the changes in productivity are primarily liked to the influx of Atlantic water or if it could be linked to wind mixing during episodes of stronger winds and/or linked to changes in runoff from land liked to precipitation, it would be relevant to also compare with precipitation data. Here, e.g., studies of mass balance in Norwegian glaciers (e.g. Nesje et al 2000) as well as lake studies would be relevant. Nesje, A., Lie, Ø. and Dahl, S.O. 2000: Is the North Atlantic Oscillation reflected in Scandinavian glacier mass balance records? Journal of Quaterny Science 15, 587-601.

Another point to raise is the actual use of the data. The dataset includes benthic foraminiferal assemblage studies, including factor analyses, planktic foraminiferal concentrations (no details on species distribution, so I assume that this was not analyses?), Mg/Ca, Mn/Ca, and stable carbon and oxygen isotopes. However, the geochemical data is only used for calculating bottom-water temperatures, and these temperatures are more or less accepted without any further discussions. The reliability and uncertainly of the data needs to be taken into account. Thus, the discussions on the palaeoproductivity is almost solely based on the benthic foraminiferal assemblages. The benthic foraminifera are good indicators, but since so much more data exist and are presented, they should also be used properly in the discussions.

Finally, despite an introduction trying to build a link between this study and the understanding of the consequences of greenhouse gas emissions, it the actual significance of the study region is not clear: Why is Skagerrak relevant? Because it represents and intermediate area between the open ocean and coastal regions? Also the actual

relevance of the outcome of the study to the problems raised in the introduction is not clear and should be made clear in the conclusions. It is an interesting and relevant study, but please make it clear to the reader, too.

Minor comments:

Latin grammar rules means that the name of the species should be "Melonis barleeanus", not "Melonis barleeanum".

Are you sure that Cassiculina neoteretis is present in the material? If yes, this could indicate an high influx of deep Atlantic or even Nordic Sea water.

Page 4, lune 25-30: It is not quite clear from the description of the water masses, which ones are surface waters, intermediate waters and bottom waters. One for one water mass is the depth in the water column provided. As the depth of outflowing/inflowing waters is very important, this must be made clear. It also needs to be specified very clearly, which water mass sweep the actual study sites in the deep Norwegian Trench.

Page 5, line 12-15: It should be pointed out specifically that both cores are taken from the deep Norwegian Trench.

Material: please provide a short, overall description of the sediment in the two cores.

Factor loadings are provided and used in the discussions and presented well in the final figure of proxy comparison. However, in order to evaluate the results of factor analyses, the authors should consider actually plotting them vs age in a diagram comparing them to the faunal data.
* * *

---

## Author Comment (AC1) · 19 Jun 2018

Response: We appreciate constructive comments and suggestions from anonymous referee 1. Responses to the review are below.

Reviewer 1: The paper presents a new data set from two high-resolution cores in the Skagerrak, spanning the last 1.1 thousand years, with the purpose of investigating historical changes in productivity. The data itself is a nice contribution to our understanding of recent changes in the ecosystem and forms a good follow-up to previous work from the same group and is very relevant to the topics covered by Biogeosciences. However, I have some concerns about the overall discussion and interpretation of the

[Figure]

data. Additional data is not needed, but the existing data should be more carefully reconsidered.

In particular, the main conclusions about changes in productivity appear to rely closely on only one part of the data set i.e. the benthic foraminifera records. The authors identify three periods of different productivity based on some good analyses of the changes in benthic assemblages, but then appear to try to match each of the other records (TOC, $\delta$13C, Mg/Ca, $\delta$18O, Mn/Ca) to these three periods, which in many cases does not seem appropriate based on the figures provided. For example, in description of the Mg/Ca results, the authors highlight high temperatures between 900 and 1200 AD, I suppose because 1200 AD is where they have identified a benthic change. However, in my opinion, for the core EMB046/10-4GC this period does not stand out, and for the core EMB046/20-3GC the temperatures seen in the earlier period continue until perhaps 1400. More careful analysis and description of this and the other records should be provided.

Response: We will clarify this point by adding sentence to the discussion as follow: 'The original subdivision of the record into intervals was done based on foraminiferal data. At the same time isotope and Mg/Ca data seem to not follow those subdivisions instead show most pronounced changes, similar for both cores, between $\sim$ CE 1650 – 1850.' We will revise result and discussion section in a way to use more objectively all given data.

Major comments:

Reviewer 1: In addition, some of the speculation regarding sources of water/nutrients to the area (particularly the Atlantic Water) needs some more thought, and could be improved by trying some additional calculations based on the existing data.

Response: We have calculated $\delta$18Osmow and have included the results in the modified fig. 4 and 6. We will describe the obtained data in the results part and use them in the discussion accordingly. Also we have added more information to the introduction about nutrients origin, see response to the comments below: 'Reviewer 1: Pg 3 Line 20'; 'Reviewer 1: Pg 3 Line 6'. In addition, we have modified the study area part of the paper by giving more details about the different water masses (please see answer to the review 2) and their origin, and we have made it clear at which levels in the water column these water masses are found. We hope that presented now the water masses description is more clear hence we will follow this description while improving the discussion part.

Reviewer 1: Uncertainties in the data need to be better characterised in the text and figures.

Response: We have added information about uncertainties of TOC and Mg/Ca to the text. Uncertainties in the data have been plotted, accordingly. Please see attached Figures.

Reviewer 1: Also, I think the authors could use the introduction to better highlight the importance of their work for a reader not familiar with the Skagerrak. Response: We have made this point clear by adding information to the introduction as follow: Pg 3, line 21: 'The North Sea and the Skagerrak absorb large quantities of atmospheric $CO_2$ via biological pomp (1.38 mol C m-2 yr-1 and 1.2 mol C m-2 yr-1, respectively), thus both play a large role in the carbon cycle (Thomas et al. 2005; Hjalmarsson et al., 2010). The Skagerrak acts as a main depositional basin for about half of the refractory carbon produced in the North Sea and for a high amount of labile organic matter imported with waters of the near-bottom current from the Danish coast or caused by intense algal blooms (Boon et al., 1999). Input of nutrients largely regulate food webs, which makes nutrients of great economic importance for the coastal areas worldwide (Micheli, 1999; FAO, 2016) and in the Skagerrak nutrients are particularly important for the Nordic fisheries (Hop et al., 1992; Iversen et al., 2002; Olsen et al., 2004; Skogen et al., 2007). Fisheries and aquaculture sectors of the Skagerrak commercially valuable for the Scandinavian nation also make a relevant contribution to meeting growing global demand for food, which will largely rely on coastal regions to host a major part of food

production in the future (e.g. FAO, 2013).' Pg 3, line 29: 'To better understand ongoing and possible future changes in productivity and related to its environmental effects more historical studies are needed'

Specific Substantial Comments Motivation Reviewer 1: As a reader unfamiliar with the Skagerrak, I felt that more was needed in the introduction to explain why the work is important and to highlight the interest of the work to a wider audience. There are references to changes in the ecosystem, but it would help the reader a lot to know more specifically why changes in production are important to the region (e.g. fisheries etc). Perhaps put this motivation up front before the more technical introduction.

Response: For the changes in the text please see response above.

Uncertainty Reviewer 1: The uncertainties on each record need to be better characterised throughout the paper. In particular, uncertainties should be quoted for all methodologies, and the level of confidence indicated. At present it is unclear whether the errors that are quoted are at 1 or 2 sigma, and this makes a huge difference. It would be helpful to have a bar showing the uncertainty beside each record in the figures as well.

Response: The errors are quoted at 1 sigma. More information has be added regarding uncertainties. See responses to specific comments underneath.

Pg 5 Line 20: What is the TOC uncertainty? Even though it is published elsewhere it wouldn't take much space to include it here. Just a sentence to say uncertainty is 0.01 % (confidence).

Response: Description of the TOC analysis have been modified as follow: 'The TOC was determined using 'Rapid CS cube –Elementar' analyser (Department of Geology and Paleogeography, University of Szczecin, Poland) with a measurement accuracy of 0.01% (95% confidence level (CL), 99.5% detection limit (DL)).'

Reviewer 1: Pg 6 Line 3: What is the level of confidence for these uncertainties? And

are they long term precision or just the precision for the runs?

Response: We have modified the sentence according to the comments as follow: 'The long-term analytical uncertainty is $\pm0.08$‰ and $\pm0.03$‰ (95% CL) for oxygen and carbon isotopes, respectively'.

Reviewer 1: Pg 6 Lines 19-27: Discussion of errors needs to be clearer in this section. Response: We have modified the sentences according to the comments. We have changed the section method for the trace element analysis by characterizing the uncertainties through the section. (modified version): "Shells of M. barleeanum were also cleaned and analysed for trace elements at the Trace Element Lab (TELab) at Uni Research Climate, Bergen (Norway). For each analysis, approximately 15-20 specimens were gently crushed between two glass plates under a microscope to allow the contaminants to be removed. The samples were cleaned following the procedure described by Barker et al. (2003). The cleaning method includes clay removal steps, oxidation of the organic matter and surface leaching. Samples containing enough material were mixed and split into two subsamples to allow duplicate analysis. All samples were dissolved in trace metal pure 0.1M HNO3 (prepared from HNO3 TraceSELECT$^{®}$) and diluted to a final concentration of 40 ppm of calcium (Ca). The trace elements were measured on an Agilent 720 inductively coupled plasma optical emission spectrometer (ICP-OES). Six standards have been prepared at TELab and have a composition to similar foraminiferal carbonate (0.50 – 7.66 mmol mol-1). Further, Fe/Ca and Al/Ca values have been checked and show no correlation with the measured Mg/Ca values. The correlation coefficients ($R^2$) between Mg/Ca and, Fe/Ca and Al/Ca ratios are respectively, for the core EMB046/20-3GC, 0.012 and 0.095, and for the core EMB046/10-4GC, 0.020 and 0.067, indicating no systematic contamination due to insufficient cleaning. The Mn/Ca values in our samples are higher than the recommended maximum (<105 $\mu$mol mol-1) (Boyle, 1983), indicating that diagenetic coatings might also affect our results. The Mn/Ca values, however, show no significant correlation with the measured Mg/Ca values ($R^2 = 0$ for EMB046/20-3GC and $R^2 = 0.021$ for EMB046/10-4GC).

Standard solution with Mg/Ca of 5.076 mmol mol-1 were analysed after every eight samples to correct for instrumental biases and analytical drift of the instrument. The long-term Mg/Ca analytical precision ($1\sigma$), based on the standard solution, is $\pm0.016$ mmol mol-1 ($1\sigma$ Standard Deviation) or 3.11% (relative SD). Average reproducibility of duplicate measurements (pooled SD, dof = 41) is equivalent to an overall average precision of 4.09%. The average Mg/Ca of long-term international limestone standard (ECRM752-1) measurements was 3.76 mmol mol-1 ($1\sigma$ = 0.07 mmol mol-1) with the average published value of 3.75 mmol mol-1 (Greaves et al., 2008). The published Melonis spp. Mg/Ca - Bottom Water Temperature (BWT) is calculated from the measured Mg/Ca using the new Melonis spp. calibration (Mg/Ca = 0.113($\pm$0.005)*BWT + 0.792($\pm$0.036)), based on core top data covering a Mg/Ca range of 0.68 – 3.66 mmol mol-1 for a temperature range of -0.89 – 15.58°C (Hasenfrantz et al., 2017). According to the calibration uncertainty, an $1\sigma$ temperature error (95% CL) of $\pm$0.9°C to $\pm$1.7°C has to be taken into consideration for the temperature range (4.1 – 9°C) covered by EMB046/20-3GC. For EMB046/10-4GC the $1\sigma$ temperature error (95% CL) is slightly similar between $\pm$ 1°C to $\pm$1.6°C for the temperature range between 5°C and 8.5°C. Further discussion on Mg/Ca-derived BWT within this article will use the Hasenfratz's calibration.' Reviewer 1: Line 19: what is the 0.016 long term precision measured on (I assume a standard material) and what is it's Mg/Ca value? What is the confidence level?

Response: The 0.016 mmol mol-1 long-term precision refers to the long-term standard deviation based on the analysis of an in-house standard solution with Mg/Ca of 5.076 mmol mol-1 after every eight samples. We have changed the sentence according to the points listed by the reviewer, please see above.

Reviewer 1: Line 20: Join these two sentences together (e.g. 'show a good reproducibility, with the pooled. . .'.

Response: We have combined the two sentences together, please see above. The new sentence reports the reproducibility and indicates the pooled standard deviation.

Reviewer 1: Line 22: This is ok but the authors still need the confidence level.

Response: The error has been quoted to 1 sigma, please see above.

Reviewer 1: Lines 25-27: I would like to see an estimate of the typical temperature uncertainty obtained when propagating Mg/Ca ratios through the calibration equation here. According to the Hasenfratz paper, the $\pm$ 2 uncertainty ranges from $\pm$ 0.6 to 2 oC depending on the temperature. I guess we might be looking at $\pm$1.5 oC, and most of the wiggles in Figure 3 might fit within these uncertainty bounds.

Response: We have added the temperature uncertainties obtained for the range of temperature we have for the two different cores. We have also modified all figures where the Mg/Ca-Temperature records are shown and include the uncertainties so that the temperature uncertainties linked to the used calibration is shown.

Reviewer 1: The authors really need to make sure that the moving average record is real and not just random scatter.

Response: The now added uncertainties to the plots show that the smoother record can be considered representative for the real long term signal. We will take that into account when revising the manuscript. In the new version the uncertainty bands are shown together with the smoothed records, showing more clearly the relationship between the raw datasets and the smoothed records.

Reviewer: Pg 6 Line 28: I am less concerned that errors on counts would be a big problem for the results, especially for those with the biggest signals. However, it would still be nice to have an estimate of the counting uncertainty for each species. Ie. Replicate a sample several times how close can you repeat the counts? There are also some more formal ways to estimate counting uncertainty. See e.g. (Heslop et al. 2011, Diagnosing the uncertainty of taxa relative abundances derived from count data, Marine Micropaleontology 79, pp 114-120).

Response: We thank the reviewer for an advice on the study by Heslop et al. (2011).

However, given the high number of samples (161 samples), which had to be analysed and at least 300 ind. counted per sample, the replicate analysis was outside of scope of this study due to a lack of time and resources. Also, for foraminifera analysis we used wet-counting method, which means that foraminifera were counted straight after washing. After that samples were dried and stored for further analyses (isotopes and Mg/Ca), which also resulted in a limited amount of material available for foraminiferal replicate analysis. The wet counting method allowed us to reduce loss of fragile agglutinated tests, thin-shelled taxa (e.g. Alabominella weddelensis, Epistomienlla spp., Nonionella iridea) or inner organic linings (IOL) (Duffield and Alve, 2014; Brodniewicz, 1965) and in consequence provides more accurate record of foraminifera assemblages. Because of the chosen wet-counting method a repeat of counts could be done only on dried material what may cause differences between the replicates due to loss of fragile agglutinated and thin-shelled forms. Therefore, uncertainty for each species cannot be delivered from the replicate counting. Also since the extreme numbers we find for some species (e.g B skagerrakensis incl its juvenile forms) are clearly discernible in both cores presented herein (plus another core off western Norway – not shown) we believe that they are not the artefact of a chosen counting method but are rather true numbers.

Reviewer: Pg 7 Line 10: It would be useful to know what the typical age uncertainty for a given depth might be for both cores. The period after 1900 looks well-constrained by Hg, but radiocarbon can be difficult to use to match two cores exactly. However, I don't think the age model should affect your results too much. My main concern would be in saying things like 'between 1550 and 1650 one core shows this while the other core shows that.' Can the cores really be associated to that level of accuracy?

Response: We agree with the reviewer that there are limits to the certainties that can be set for radiocarbon dated chronologies' that influences the interpretation of changes at short time scale (e.g. multidecadal). We will modify the text accordingly to the comments.

Furthermore, since submitting, we have realized that there might have been some reason for concern regarding some of the dates used to set the chronology and have therefore re-dated several levels and created new age models, to improve the confidence that can be put on the age model, important when investigating such short time scales. We have also changed the $\Delta R$ value used for calibrating the dates from $0\pm50$ to $200\pm50$, since this improves the relationship between the historical tie points and radio carbon dates. When establishing the original age model, the two cores were set at a common depth scale. We keep this correlation after updating the absolute ages. Hence, the relationship between the two cores has not changed, but potential changes in the relationship comparing to other records will be re-evaluated. The new age model will be presented in full in the revised manuscript.

Describing the records in terms of the three periods Reviewer 1: This was my main area of concern when reading through the paper. Firstly, I note that the three time periods loosely correspond to different sampling resolutions (i.e. relatively high resolution for the early and late periods and relatively low resolution for the middle period). Could this be affecting where lines were drawn? I don't think it is a big problem but it could be worth some investigation or a note in the text.

Response: We double checked sampling resolution. The core EMB046/20-3GC was analysed for foraminifera every $1 - 3$ cm with an exception of 7 cm interval between $89 - 96$ cm ($\sim$ CE 1055 – 1030 years). The core EMB046/10-4GC was counted every $1 - 2$ cm. The timing of the periods as set did not correspond to times when the sedimentation rate changed according to the age model. Thus, the different distribution of foraminifera data versus age scale is caused not by resolution differences (those are relatively steady). Originally, there were indicated a strong increase in sedimentation rates for the oldest part of the record. Using the improved age model, this is no longer the case. The new age model shows more realistic changes relative to what is common in this area during this time. As mentioned above, the full new chronological framework will be presented in the revised manuscript.

The method part will be modified as follow: (previous version): "The two Skagerrak records over the targeted time interval covering the last 1100 years were counted at 1 – 5 cm and 1 – 2 cm resolution for EMB046/20-3GC and EMB046/10-4GC, respectively" (modified version): "The two Skagerrak records over the targeted time interval covering the last 1100 years were counted at 1 – 3 cm with an exception of 7 cm interval between 89 cm and 96 cm ($\sim$ CE 1030 – 1055) for EMB046/20-3GC, and 1 – 2 cm resolution for the entire 1100 years of the EMB046/10-4GC record".

Neither is the timing of the periods influenced by changes in sedimentation rates, as these change within the periods are not at the transition.

Reviewer 1: Secondly, as I noted above, the most convincing case for separating the record into three periods seems to be the faunal data. If I were to ignore those data and to try to describe the temperature, TOC, $\delta$13C and Mn/Ca I would not find the patterns described in the text. The description of the temperature signal does not seem very objective. I gave one example above, and I note again how the record for EMB046/10-4GC doesn't seem to have any period that is very different, especially once you include the temperature uncertainty. I can see a period of low temperature for the other core between maybe 1450 and 1650-1700, but I do not see why we would distinguish the first 200 years from the next 500. The temperature and oxygen isotope records need a more thorough and objective analysis of whether they do actually match the faunal data, especially as the authors use such a match to argue for changes in water mass driving the faunal changes.

Response: We believe that foraminifera are good indicators for productivity changes however we agree that other data presented in our study should be discussed in more detail and even though they do not necessary follow the foraminiferal pattern they should be examined in terms of different forcing factors, which might influence changes of productivity e.g. changes in water masses, nutrients sources, temperature variability. Therefore, we will revise the discussion part in a way where all given data are paid equal attention.

Reviewer 1: On Pg 12, Line 20 the authors describe the TOC as low, during the period of low productivity. However, it is higher than in the first period (argued to be moderately productive) and where the authors have also argued for lower oxygen concentration based on Mn/Ca. I find it hard to see how you could have less production, more bottom water oxygen and yet higher TOC if TOC is being considered as a productivity proxy.

Response: We agree with the reviewer that TOC records does not show any major changes in the interval between ∼ CE 900 – 1600. The most visible change in the TOC records appear after ∼ CE 1700 (according to the new age model). We have removed 'accompanied by low TOC' from the sentence 'The low absolute abundance of planktonic foraminifera and benthic palaeoproductivity fauna accompanied by low TOC between ∼ CE 1200 and 1600 indicate lower primary productivity than between ∼ CE 900 and 1200 (Fig. 4).' and we have added information about TOC into the sentence on pg. 12, line 26 as follow: 'The $\delta$13C and TOC values seen during the interval ∼ CE 1200 to 1600 are overall comparable to the previous interval (∼ CE 900 – 1200) (Fig. 4).' We decided to not include the Mn/Ca data into the manuscript, since the Mn/Ca presented on new age model does not show any significant changes.

Reviewer 1: How about the possibility that the sediment TOC gets less down the core simply because it is decaying over time?

Response: We agree that the decrease in TOCs with depth at both sites is probably in part caused by decaying organic matter. That is why we try do not address TOC changes in details for the part of the record older than CE 1700. However, the pronounced increase in TOCs after ∼ CE 1700 towards cores top coincides with changes in foraminiferal assemblages as higher abundance in palaeoproductivity fauna and increases in B. skagerraknesis factor suggesting higher productivity at that time. Higher productivity led to increase in organic matter in the sediments and most likely also contributed to changes in TOC profile.

Influence of different water masses

Reviewer 1: Given the issues I have outlined above, the inferences that different water masses are influencing the sites at different time (e.g. Atlantic Water contributing to moderate productivity during the early period), are not well-supported. For example, the authors argue that warmer conditions (from Mg/Ca and $\delta$18O) during the 900-1200 period can be explained by increased Atlantic Water. Firstly, I suggest that the authors need to robustly test their trends with appropriate errors as outlined above.

Response: We have modified Figs. 4, 5, 6 by adding error bands to the BWT-drived Mg/Ca and $\delta$18O.

Reviewer 1: Secondly, the arguments based on temperature (especially in this section, Pg 11) are split by a section on carbon isotopes, which confuses the flow of the paper. There seem to be two arguments for temperature change: increased Atlantic Water influence and generally high temperature during the MWP. I think the authors do combine these arguments by reference to NAO changes, but the link is not made very clear. In addition, the authors should at least address why the temperature in core EMB046/20-3GC remains high well into their second time period, when the productivity has already decreased, which they overall argue is a partly a response to less AW.

Response: We will make this point clear by restructuring discussion part following already modified study area part. We also believe that added new figure (Fig. 6) where we compared our data to the NAO records reported by Trouet et al. (2009), Olsen et al. (2012), Faust et al. (2016) will help to clarify connection between water masses changes, NAO, temperature and climate. We will revise the section such that the discussion about carbon isotopes no longer disturbs the flow.

Reviewer 1: There is some discussion of complicating factors that might affect the temperature/productivity link, but it is not very clear and should be restructured. To solidify the link to increased AW I would suggest using the Mg/Ca and $\delta$18O to reconstruct the $\delta$18O of the deep water (e.g. salinity), because the $\delta$18O of the shells also has a temperature signal. Knowing the water $\delta$18O might provide a more useful indicator of

water provenance and is standard practice in most paleoceanographic work, although the uncertainty might again be very high in this case. The references used to support the conclusions here seem appropriate, and they may be the simplest and best way to argue that the changes seen are do in part to water masses.

Response: We thank reviewer for the suggestion. We have calculated $\delta$18Osmow and have included the results in the fig. 4 and 6. We will describe the obtained data in the results part and use them in the discussion accordingly.

$\delta$13C records Reviewer 1: The authors attempt to put the $\delta$13C records into the context of their productivity results, and they note correctly that the late increase in $\delta$13C is due to the Suess effect. However, given that changes in water masses are invoked to explain changes in productivity, I would expect to see some analysis regarding whether their $\delta$13C changes could be explained this way, rather than by changing productivity at the sites. I think the authors correctly conclude that the $\delta$13C records do not correspond to the productivity records. Therefore, I would recommend removing much of the $\delta$13C discussion, simplifying it to a sentence or two saying that not much coherence is seen. I also think the Suess effect is well known enough that such a long paragraph from Pg 16 line 10 is not really warranted. Cutting out the $\delta$13C discussion would also help streamline the paper.

Response: We will modify this paragraph accordingly to the comments.

Specific Minor Comments

Reviewer 1: Pg 3 Line 6: Note that in general, not all organic matter makes it to the seafloor (in fact most doesn't), and much of it is remineralised in the water column. If this situation is different in the Skagerrak the authors should say so and why.

Response: The Skagerrak forms the deepest part of the Norwegian Trench and therefore acts as a sink for organic matter produced in the North Sea. Boon et al. (1999) presented an interesting study on distribution, metabolism of organic matter and variation in the supply of fresh phytodetritus to the sea floor in the Skagerrak and neigh-bouring area. They reported that the Skagerrak is a decomposition area for about half of the refractory carbon produced in the North Sea but also for a high amount of labile organic matter. The high amount of fresh organic matter found in the sediments is likely caused by intense algal blooms or advection of labile organic matter from the shallower coastal waters of northern Denemark transported via near-bottom currents to the Sk-agerrak (Boon et al. 1999). Because the Skagerrak consist of fine-grained silty-clay sediments it is possible that mineralisation of labile organic matter slows down due to sorption to mineral surfaces of sediment particles (Keil et al. 1994). We have added more information about organic matter in the Skagerrak accordingly to the comments and the response given above as follow: Pg 3 line 21: The Skagerrak acts as a main depositional basin for about half of the refractory carbon produced in the North Sea and for a high amount of labile organic matter imported with waters of the near-bottom current from the Danish coast or caused by intense algal blooms (Boon et al., 1999).

Reviewer 1: Pg 3 Line 9: Sentence reads a little strangely. Consider rephrasing to something like 'Via photosynthesis, primary producers can help to remove CO2 from the atmosphere as part of the biological pump'. Response: Changed accordingly. Reviewer 1: Pg 3 Line 11: Change to 'likely negatively impact'. Response: Changed Reviewer 1: Pg 3 Line 13: Sentence needs rephrasing. Need to specify 'export of organic matter' rather than 'production'. Response: Changed Reviewer 1: Pg 3 Line 18: Include Figure 1 reference here to guide readers who don't know anything about the Skagerrak. Response: Added Reviewer 1: Pg 3 Line 20: How does increased air-sea gas exchange lead to increased nutrients? Perhaps expand a little. Response: We have added 'atmospheric CO2 uptake' to the sentence Pg 3 line 16: 'Coastal zones are among the most productive marine regions characterised by high: atmospheric CO2 uptake, organic matter accumulation and decomposition (e.g. Hjalmarsson et al., 2010)'. We have also added the fallowing sentence: 'The North Sea and the Skagerrak absorb large quantities of atmospheric CO2 via biological pump (1.38 mol C m-2 yr-1 and 1.2 mol C m-2 yr-1, respectively), thus both play a large role in the carbon cycle

(Thomas et al. 2005; Hjalmarsson et al., 2010)'. We believe that these changes and also first paragraph with more technical introduction sufficiently addressed the air-sea gas exchange topic. Reviewer 1: Pg 3 Line 23: 'positive impacts on growth'. Growth rate? Growth magnitude? Be more careful about saying 'positive impacts'. Do you mean positive in the sense of being 'good', or simply 'more'? Response: 'positive impacts on growth' changed to 'positive impacts on growth rate" Reviewer 1: Pg 3 Line 24: Similar to above, what is a negative change in trophic levels? Fewer trophic levels? This needs to be more specific. Response: 'negative' changed to 'disruptive' Reviewer 1: Pg 4 Line 7: At the first introduction of the Kattegat, provide reference to Figure 1. Response: Added Reviewer 1: Pg 4 Line 9: It might be better to say that 'in the past the NAO has represented one of the leading modes of natural climate variability over the North Atlantic'. Or remove the word 'natural' altogether. I think it may now be difficult to say whether the NAO is wholly 'natural' anymore. Response: We removed 'natural' - pg 4 line 7 and 'naturally' - pg 4 line 1. Reviewer 1: Pg 4 Line 12: Specify what you mean by BCE. Response: We added 'Before the Common Era (BCE)'. Reviewer 1: Pg 4 Line 26: 'governed by' could be more specific. How does the sill control the AW? Response: We have modified study area part. Please see comment above. Reviewer 1: Pg 5 Line 2: 'internal' might not be the best word here. Internal to what? Perhaps say 'In addition to the mean circulation, processes such as: : :' Also as a general point, it helps the reader if all the references can be at the ends of sentences where possible. It is possible in this instance. Response: We have modified study area part. Please see comment above. We will try to keep all the references at the ends of sentences where possible. Reviewer 1: Pg 5 Line 16: There are two sentences here that are almost the same. Consider rephrasing to avoid repetition. Response: We have modified these sentences as follow: (modified version): 'This study based on results from the upper 170.5 and 164.5 cm, in cores EMB046/20-3GC and EMB046/10-4GC respectively, what correspond to the last 1100 years. Here we present new stable isotope ($\delta$13C, $\delta$18O), and trace element ratios (Mg/Ca, Mn/Ca) data covering the last 1100 years, in combination with foraminiferal assemblage data

and multivariate statistics.'

Reviewer 1: Pg 5 Line 25: Avoid the use of 'proved' (or proven). I think it is fine to just say 'potential'. Response: We changed this through the manuscript. Reviewer 1: Pg 6 Line 1: What grade of methanol was used? Response: The grade of methanol is $\geq$ 98.8 %. We have added this information to the text. Reviewer 1: Pg 6 Line 1: 'For each measurement 2-4 specimens were used.' How many measurements were done for each depth? Are 2-4 specimens enough to get a robust average isotope signal? Response: The isotope measurements were run in high resolution (1 cm intervals), no replicates were conducted. For each measurement 50–100 $\mu$g was needed which is equal to ca. 2–4 M. barleeanus specimens. Such an amount of tests is sufficient for a sensitive mass spectrometer (Finnigan MAT 253) at Bergen University and has been previously used by many others (e.g. Milzer et al. 2013), thus we believe that by following the known procedure for isotope analyses, we were able to obtain robust isotope signal. Based on experience from the area we think that we get a robust isotope signal even though few specimens are used in each analysis. We do however focus on the trends, not the results of individual measurements as we agree that relying on individual measurements might overestimate the accuracy of the information we can extract. Reviewer 1: Pg 6 Line 10: I think the authors could be more specific about the grade of HNO3 used. What Ca concentration were the final solutions? Were these matched to the standard concentrations? Response: We have rewritten the sentences according to these comments. Please see above the modified 'trace elements' paragraph. Reviewer 1: Pg 6 Line 14: These results do not indicate 'no contamination', only 'no systematic contamination'. The authors are correct in the following sentence where they indicate that because of their high Mn/Ca values, contamination may be an issue. No need for the sentence on Line 16, as it repeats what was already said. Response: According to points listed, we have modified the sentences. Please see above the modified 'trace elements' paragraph. The sentence on Line 16 has been deleted. Reviewer 1: Pg 6 Line 18: 'After every 8 samples'? 'For' might not be specific enough. Response: We have changed the sentence according to this comment.

Please see above the modified 'trace elements' paragraph. Reviewer 1: Pg 6 Line 18: Which standard solutions were used? Response: We have added a sentence presenting the Mg/Ca concentration range used for the standard solutions. Please see above the modified 'trace elements' paragraph. Reviewer 1: Pg 6 Line 28: 'The foraminiferal faunal analysis' Response: Added Reviewer 1: Pg 7 Line 14: Include 'IntCal' in the wording somewhere here. Response: Added Reviewer 1: Pg 7 Line 21: What is the detection limit for the TOC measurements? I think the results look fine but might be worth checking. Response: The detection limit for the TOC measurements is larger than 99.5 %. We have added this information to the text. Reviewer 1: Pg 7 Line 26: Here is an instance where knowing what the relative age uncertainty when comparing the different cores could be important. 1550-1650 is not a very long time period in terms of typical radiocarbon uncertainties. Response: The age uncertainty of the carbon dates in our records vary between 30 and 60 years. Thus, we agree with the reviewer that interpretation of changes in short time periods (e.g. multidecadal) cannot be well verified. We will modify the text accordingly to these comments. Reviewer 1: Pg 7 Line 28: To me 'long-term trends' implies a gradual change to different conditions over the whole record. Instead these records are quite flat until the Suess effect kicks in. I might say 'long-term variations'. Response: Changed to 'long-term variations'. Reviewer 1: Pg 8 Line 6: Same comment as above but now for oxygen isotopes. Response: Changed to 'long-term variations'. Reviewer 1: Pg 8 Line 26: I know that identifying planktonic forams is very difficult at the tiny sizes looked at here. I agree with the author's decision to discuss only the total number rather than species. Do the benthic ID's suffer from the same difficulty. I have less benthic experience to offer a full comment here. Response: There are not similar issues regarding benthic foraminifera since most of their juveniles are easily discernible at the species level. We observed a wide range of sizes in benthic taxa, from very small e.g. Alabaminella weddelensis, Epistominella sp. (E. exigua, E. vitrea), Nonionella iridea or Cassidulina sp. (C. laevigata, C. neoteretis) to bigger e.g. Melonis barleeanus, Pullenia bulloides, Hyalinea balthica. The advantage for the identification of the foraminiferal assemblage from the

Skagerrak region is that many research studies performed before have targeted this topic and as a result there is a lot published information about benthic foraminiferal taxonomy from the region.

Reviewer 1: Pg 9 Line 7: What is meant by 'less fluctuating'? Apart from one very large peak in 20-3GC the records look quite similar. Are the standard deviations very different? Response: We modified this sentence accordingly to the comment. (modified version): 'The benthic foraminiferal record from the core EMB046/10-4GC is characterised by consistently high absolute abundances (123 – 455 ind. g-1 wet sed.) in contrast to overall lower values in the core EMB046/20-3GC (43 – 527 ind. g-1 wet sed., where the highest value represents an individual peak above 361 ind. g-1 wet sed. significantly standing out from the rest of the record).' Unfortunately, we cannot give standard deviations (please see comment above concerning the uncertainty in foraminiferal records). We have calculated standard deviations (SD) for entire records (total benthic forams g-1) for each core. For EMB046/20-3GC the SD is 79.92 and for EMB046/10-4GC the SD is 75.82.

Reviewer 1: Pg 9 Line 9: How does changes in mass accumulation affect the results? When looking at forams per gram, two things can change the magnitude: 1) number of forams and 2) grams of sediment. Can the authors rule out changes in mass accumulation rate with their age model? Response: Unfortunately, we do not have sediment volume data to calculate mass accumulation. Reviewer 1: Pg 9 Line 18: This statement seems quite broad. Several of the low abundance benthics do not seem to show any trends (perhaps because of the counting uncertainty?). Some of them show different trends. I think B. skagerrakensis can be highlight more here, because it is really the main record that shows any large and consistent changes.

Response: We modified this sentence according to the comment. (previous version): 'In general for both cores, the relative and the absolute abundance of the benthic foraminifera follow similar trends, showing a distinct variability of species B. skagerrakensis (Fig. 2). (modified version):'Among benthic foraminiferal species Brizalina

skagerrakensis shows the most prominent and consistent changes when comparing both records (Fig. 2).'

Reviewer 1: Pg 10 Line 6: This first sentence needs rephrasing. What is meant by 'quality'? Again avoid 'proven'. Is it the individual species or the 'factors' that have an association with organic matter? I don't think the wording makes it clear. Response: We have modified this sentence accordingly to the comments. The individual species have an association with organic matter thus factors related to food preferences of species are included in each factor. (previous version): 'All dominant species in our benthic foraminiferal assemblages, grouped into factors, have proven association with quality and availability of organic matter at the sea floor (...).' (modified version): 'All dominant species in our benthic foraminiferal assemblages, grouped into factors, have documented association with quality (e.g. fresh or decaying) and availability of organic matter at the sea floor (...).' Reviewer 1: Pg 10 Line 15: The following 3 sections should be rewritten taking into account the substantial comments above, taking more care over the uncertainties in each record, and appreciating more that e.g. temperature and oxygen isotopes don't necessarily match the benthic productivity records. Abstract and conclusion: Can be looked at again after more careful consideration of the data. Response: The discussion section will be revised according to the comments. We believe that added now uncertainties to the data will help with clarifying the discussion of the results. Also we will take into account the other data presented in the manuscript (Mg/Ca, isotopes, now added $\delta18Osmow$) and discuss them properly. Reviewer 1: Figure 2: n g-1 I assume to mean number per gram. It could be misunderstood as 'per nanogram' so perhaps change to 'Number g-1'. Response: 'n g-1' has been changed to 'ind. g-1' in all figures and through the text.

Reviewer 1: Figure 4: For the uninitiated like me, 'Factor loading' doesn't mean very much as an axis title. Perhaps explain a bit more in the text. Response: We have clarified this point by adding sentences to the results section (Pg 9, line 31), as follow: 'The weight (importance) of each factor found in the sample is expressed by each factor

loading (Fig. 4). Factors with loadings above 0.5 are considered most significant.'

Technical Corrections Reviewer 1: There are a large number of instances where the language could be tightened up in the paper that would help make it easier to read. I would suggest having the paper thoroughly proof read again. Previous papers from some of the same authors read well. I have put a very few points below. Response: The manuscript will be tweaked, the language tightened up and references double checked. Reviewer 1: Pg 5 Line 29: Change 'since' to 'because'. I have had it pointed out to me before that 'since' can be a confusing word in a paper dealing with the past. Response: Changed Reviewer 1: Pg 6 Line 12: Rephrase sentence to 'Fe/Ca and Mn/Ca ratios show no correlation with measured Mg/Ca values.' Response: Changed Reviewer 1: Pg 6 Line 26: Hasenfratz miss-spelled. Response: Corrected Reviewer 1: Pg 9 Line 8: Change 'sticking out' to something sounding more scientific. Response: Changed to 'standing out' Reviewer 1: Figure 3 caption: '5-point' not '5-points'. Response: Changed

[Figure]

[Figure]

Fig. 3

Fig. 1.

[Figure]

Fig. 4

**Fig. 2.**

Fig.5

[Figure]

**Fig. 3.**

[Figure]

Fig. 4.

Figure capture

Fig. 2 Age model description will be presented in the new version of the manuscript.

Fig. 3 Foraminiferal assemblages including dominant and accessory benthic species for both cores (EMB046/10-4GC and EMB046/20-3GC), and CABFAC results. Absolute abundance is shown as grey filed, while relative abundance as black curve with symbols. The absolute abundance of total benthic foraminifera is a sum of all species: agglutinated (Agglut.) and calcareous (Calc.). Dashed lines divide record into 3 periods of most pronounced palaeoproductivity changes, which are discussed in the text.

Fig. 4 (A) Oxygen stable isotopes ($\delta^{18}$O); (B) the Mg/Ca-derived bottom water temperature (Mg/Ca-derived BWT); (C) the stable oxygen composition ($\delta$w), of both studied cores against age. The errors bands represent uncertainties of the records. (D) Figures include data from depths at which data of all proxies ($\delta$18O, Mg/Ca-derived BWT, $\delta$w) are available (symbols). The curves correspond to 5-points running average. Grey box indicates range of instrumentally recorded temperatures of the time period between 2009 and 1924 years from the area between 57°.17–58°´N and 8°–9°.79´E at 300 – 340 dbar (ICES 2010).

Fig. 5 Comparison of absolute abundances of planktonic foraminifera, total organic carbon (TOC), stable carbon isotope ($\delta$13C), absolute abundance of palaeoproductivity fauna, CABFAC results, absolute abundance of total benthic foraminifera, Mg/Ca-derived bottom water temperature (Mg/Ca-derived BWT), oxygen stable isotope ($\delta$18O) between two studied sediment cores EMB046/10-4GC (green curves) and EMB046/20-3GC (black curves). The thicker curves correspond to 5-points running average. The errors bands represent uncertainties of the records. Dashed lines divide record into 3 periods of most pronounced palaeoproductivity changes, which are discussed in the text.

Fig. 6 Comparison of absolute abundances of planktonic foraminifera, total organic carbon (TOC), stable carbon isotope ($\delta$13C), absolute abundance of palaeoproductivity fauna, CABFAC results, absolute abundance of total benthic foraminifera, the stable oxygen composition ($\delta$w), Mg/Ca-derived bottom water temperature (Mg/Ca-derived BWT), oxygen stable isotope ($\delta$18O) between two studied sediment cores EMB046/10-4GC (green curves) and EMB046/20-3GC (black curves). The reconstructions of the NAO index. The thicker curves correspond to 5-points running average. The errors bands represent uncertainties of the records. Dashed lines divide record into 3 periods of most pronounced palaeoproductivity changes, which are discussed in the text.

**Fig. 5.**

---

## Author Comment (AC2) · 19 Jun 2018

Response: We appreciate constructive comments and suggestions from anonymous referee 2. Responses to the review are below.

Reviewer 2: The paper presents two records of benthic foraminiferal assemblages and geochemical analyses from the northern deep Skagerrak region. The records are excellent, presenting high-resolution data for the last millennium. The strength of the paper is especially the fact that data from two neighboring cores are presented, as provides evidence of a general pattern. The paper is overall well-written and clearly presented and I believe that it fits well within the scope of the journal.

[Figure]

However, the ms has a tendency to focus too much on local conditions and comparison to relatively few previous studies. It would therefore benefit from including information from a broader range of study sites as well as from other types of records, including terrestrial and lacustrine records. Also more direct comparisons between records as well as an improved precision of the discussion of water masses is needed. Finally, not all data (e.g. Mg/Ca) are actually used to any significant extend in the discussions:

Response: The concerns about the discussion and interpretation of the data are addressed. We will add broader information and more details to the discussion about water masses and Mg/Ca (also see further information below).

Reviewer 2: A key element of the paper is the link between the record and the North Atlantic Oscillation (NAO). A number of studies have suggested a more positive phase of the NAO during the MCA and a negative phase during the LIA. However, the here the authors only refer to one study without taking into account that other, earlier, studies have also made this suggestion (e.g. from off Portugal, in the Labrador sea/West+East Greenland etc.).

Response: The sentence: "In contrast, the North Atlantic Oscillation (NAO) reconstruction by Trouet et al. (2009) suggests a tendency for prevailing positive NAO conditions during the MCA and hence, south-westerlies dominating the hydrographic and meteorological regimes during winter." have been changed to: "In contrast, the North Atlantic Oscillation (NAO) reconstructions by Trouet et al. (2009), Olsen et al. (2012), Faust et al. (2016), among others, all suggest a tendency for prevailing positive NAO conditions during the MCA and hence, south-westerlies dominating the hydrographic and meteorological regimes during winter." We will add more references concerning the NAO through the text, accordingly.

Reviewer 2: Also, since the present manuscript provides a high-resolution records, these data should in fact be plotted vs. the high-resolution NAO reconstructions (Trouet et al, as well as vs. other high-resolution records such as those by Olsen et al 2012

and Faust et al 2016).

Response: We would like to thank the reviewer for the advice on papers by Olsen et al. (2012); Faust et al. 2016) and Nejse et al. 2000. We have included information and data from these studies in the revised manuscript, presenting direct comparison between those records and ours. We will also include reference to other relevant studies e.g. Bakke et al. (2008). We have plotted three suggested NAO reconstructions vs. our data. Please find enclosed Fig. 6.

Reviewer 2: It would be very interesting to see, if the overall quite well-known trend of positive NAO during the MCA and negative NAO during the LIA is also seen at shorter, decadal/multidecadal time scales. Whether such a correlation between productivity and NAO cannot be verified, it would be valuable information. Olsen, J., Anderson, N.J, Knudsen, M.F, 2012: Variability of the North Atlantic Oscillation over the past 5,200 years. Nature Geoscience 5, 808–812, doi:10.1038/ngeo1589 Faust, J.C., Fabian, K., Milzer, G., Giraudeau, J., Knies, J., 2016: Norwegian fjord sediments reveal NAO related winter temperature and precipitation changes of the past 2800 years. Earth and Planetary Science Letters 435, 84-93.

Response: While the long trend of positive NAO during the MCA and more negative NAO during the LIA correlated well with the defined in this ms productivity periods (best seen when correlated with NAO reconstruction by Faust et al. 2016, Fig.6), the correlation at the shorter time scale is harder to address. This is due to the dating uncertainties of the different records e.g. the uncertainty of the 14C dates in our records vary between 30 and 60 years, in records by Olsen et al (2012) from ±5 to ±35. Also, the relation between NAO and productivity at the decadal/multidecadal time scale could be better justified by looking at the high resolution data from the short cores (MUCs) as well as instrumental records covering last ∼120 years and comparing them to decadal trends in the NAO reported by e.g. Hurrell (1995). We intend to show the NAO-productivity correlation at the decadal time scale in the next manuscript which is currently being prepared by Binczewska et al.

Reviewer 2: The study region is influence by several different water masses, including local outflow of low-salinity water from riverine outflow, saline Atlantic water and more intermediate salinity water as a mixture of North Sea and riverine waters (e.g. the Jutland Current). However, in the presentation of the water masses, it is not always clear at which levels in the water column these water masses are found, nor whether they also influence the actual study sites. This problem continues throughout the discussion and the one gets the impression that either there is increased Atlantic water or increased low-salinity water. However, stratification could allow both. Thus, the discussion need to be much more precise.

Response: We have modified the 'study area' as follow: "The Skagerrak is located in the northeastern part of the North Sea, connected to the Baltic Sea through the Kattegat (Fig. 1). The basin has a mean water depth of 210 m and a sill depth of 270 m. With a maximum depth of 700 m the Skagerrak represents the deepest part of the Norwegian Trench (Rodhe, 1996). The area is characterised by an anticlockwise circulation and complex hydrography. The surface circulation (<30 m) is to a large extend dominated by a surface current consisting of inflowing saline water from southern North Sea and the North Atlantic and outflowing less saline water from the Baltic Sea (Danielssen et al., 1997). The inflowing nutrient-rich surface water flows along the Danish coast driven by the Southern Jutland Current (SJC) and the Northern Jutland Current (NJC) while the outflowing Baltic Sea water (BW ∼20 – 30 psu) flows as the Baltic Current (BC) along the Swedish west coast towards the northeast Skagerrak where it merges with the NJC and turns to the northwest as low saline Norwegian Coastal Currant (NCC) (Rodhe, 1996; Rydberg et al., 1996). The water flowing from the Skagerrak towards the Norwegian Sea (NCC) partly recirculates to the western Skagerrak (Rodhe, 1996). The surface water has a high nutrient concentration mostly due to the freshwater input via rivers draining from the Norwegian south coast, German and Danish east coasts, and the Baltic catchment area but also the upwelling of the underlying nutrient-rich Atlantic water is considered to be an additional nutrient supply (Gustafsson and Stigebrandt, 1996; Rodhe, 1996). As a consequence of the mixing

of different water types and the high freshwater input enhanced by precipitation, the upper layer of the surface water has low salinity (25 – 32 psu) and is determined as the Skagerrak Coastal Water (4.5 – 10 °C). The water below, the intermediate layer (30 – 270 m) is referred to as Skagerrak Water (32 – 35 psu, 4.5 – 10 °C). The Skagerrak Water is driven by the subsurface circulation (Andersson 1996). The deep water layer below sill depth (>270 m) is dominated by Atlantic Water and is recognised as the Skagerrak Basin Water (>35 psu, 5.5 – 6.5 °C) (Aure and Dahl 1994). The subsurface circulation (below 30 m water depth), consists of nutrient-rich Atlantic deep water (AW >35 psu, 5.5 – 8.5 °C) flowing though the norther North Sea, and the water from the central and southern North Sea (NSW ∼31 – 35 psu) (Rodhe 1996). The inflowing water follows the southern side of the Norwegian Trench to enter the Skagerrak in its central part, where can be mixed with fresh surface-water and flows out as the NCC (Winther and Johannessen 2006). Large-scale atmospheric systems and regional meteorological factors (e.g. precipitation and storms) influence the flow regime creating a high-dynamic system in the upper layer of the water column where water mixing is largely caused by the southwesterly winds (Gustafsson and Stigebrandt, 1996). At the same time, calmer hydrographic conditions are typical for the intermediate layer and the deep water down to ∼ 400 m with a maximum water residence time of 3 months (Andersson, 1996). This is in contrast to the renewal of the deeper water below ∼ 400 m which occurs every 1 to 3 years depending on the strength of the Atlantic water inflows (Aure and Dahl, 1994; Rodhe 1996), which closely correlate with the NAO index (Brückner and Mackensen, 2006)."

We hope that presented above the water masses description is more clear hence we will follow this description while improving the discussion section.

Reviewer 2: In this context, I do agree that the increased flux of planktic foraminifera indicated increased inflow of Atlantic water. However, how sure are the authors that the planktic foraminifera are in fact locally produced and not brought in from the Atlantic via the currents? The planktic foraminifera may not be autochthonous and even if they are,

they would likely not represent direct surface waters. Thus, this inflow may not have occurred right at the surface, but rather as a subsurface current, thus still allowing an increased surface-outflow of lower-salinity waters.

Response 2: In the foraminiferal assemblage we found mostly large test sizes of G. bulloides and N. pachyderma and small tests of G. glutinata and G. uvula. This wide range of size (adults and juveniles) suggests that the planktonic foraminifera found in the Skagerrak cores were living in nearby water masses (Murray; 1976). Because waters with suspended sediment are not favourable for the planktonic foraminifera (which is also seen from their relatively low absolute abundance), it is possible that planktonic foraminifera were floating below the surface water level. Moreover, the depth habitats in the water column of planktonic species found in our record was reported by e.g. Jonkers et al. (2010) and Schiebel et al. (2017). According to those studies N. pahyderma dwells at 50 – 100 m of the water column and has the maximum calcification at 100 – 200 m, while G. bulloides dominates waters above thermocline at < 60 m, G. uvula occurs in highest amount in the surface waters and G. glutinata prefers the upper 50 m of the water column.

Murray, J. W. 1976: A method of determining proximity of marginal seas to an ocean. Marine Geology, 22: 103-119. Jonkers et al. 2010: Seasonal stratification, shell flux, and oxygen isotope dynamics of left-coiling N. pachyderma and T. quinqueloba in the western subpolar North Atlantic, Paleoceanography, 25, PA2204, doi:10.1002/palo.20018, 2 Schiebel et al. 2017: Modern planktic foraminifers in the high-latitude ocean, Marine Micropaleontology 136, 1–13

Reviewer 2: In the discussions on whether the changes in productivity are primarily liked to the influx of Atlantic water or if it could be linked to wind mixing during episodes of stronger winds and/or linked to changes in runoff from land liked to precipitation, it would be relevant to also compare with precipitation data. Here, e.g., studies of mass balance in Norwegian glaciers (e.g. Nesje et al 2000) as well as lake studies would be relevant. Nesje, A., Lie, Ø. And Dahl, S.O. 2000: Is the North Atlantic Oscillation reflected in Scandinavian glacier mass balance records? Journal of Quaternary Science 15, 587-601.

Response: We thank the reviewer for an advice on study by Nesje et al. (2000). We have plotted productivity data versus winter precipitation (Nesje et al; 2000), however the resolution of those data is too low over the last 1100 years to add significantly to the discussion. However, we will instead do a comparison to winter precipitation data from Bakke et al. 2008, where it looks like the precipitation has an overall decreasing trend throughout the last 1100 years. We will expand the discussion with presented in Fig. 6 precipitation information, accordingly.

Reviewer 2: Another point to raise is the actual use of the data. The dataset includes benthic foraminiferal assemblage studies, including factor analyses, planktic foraminiferal concentrations (no details on species distribution, so I assume that this was not analyses?), Mg/Ca, Mn/Ca, and stable carbon and oxygen isotopes.

Response: We found the same planktonic species at both studied locations. We did identify planktonic foraminifera to species level in each analysed sample. However due to their overall low abundance we prefer to present assemblage as total number of planktonic individuals. In line with both reviewer 1 and 2 comments we will use more of the data more actively in the revised discussion.

Reviewer 2: However, the geochemical data is only used for calculating bottom-water temperatures, and these temperatures are more or less accepted without any further discussions. The reliability and uncertainy of the data needs to be taken into account. Thus, the discussions on the palaeoproductivity is almost solely based on the benthic foraminiferal assemblages. The benthic foraminifera are good indicators, but since so much more data exist and are presented, they should also be used properly in the discussions.

Response: The discussion section will be revised according to the comments. In line with suggestions from Reviewer 1 we have added the uncertainties to the data and
information on uncertainties, and thereby we will consider to what degree we can be confident in how we interpret the data. In the revised version we will also use the other records presented (Mg/Ca, isotopes, now calculated and added $\delta18Osmow$) more actively in the discussion.

Reviewer 2: Finally, despite an introduction trying to build a link between this study and the understanding of the consequences of greenhouse gas emissions, it the actual significance of the study region is not clear: Why is Skagerrak relevant? Because it represents and intermediate area between the open ocean and coastal regions?

Response: We have made this point clear by adding information to the introduction and modifying the paragraph on Pg 3, line 16 – 29, as follows: 'Coastal zones are among the most productive marine regions characterised by high: atmospheric $CO_2$ uptake, organic matter accumulation and decomposition (e.g. Hjalmarsson et al., 2010). The Skagerrak, located between the North Sea and the Baltic Sea and in the close proximity to land, has many potential nutrient sources, such as the North Atlantic, Baltic Sea, North Sea, as well as continental discharge and river runoff (Aure and Dahl, 1994; Andersson, 1996; Gustafsson and Stigebrandt, 1996). Upwelling and precipitation further increase the nutrient supply to the surface waters, additionally stimulating productivity in this region (Pingree et al., 1982; Aure and Dahl, 1994; Fonselius, 1996). The North Sea and the Skagerrak absorb large quantities of atmospheric $CO_2$ via the biological pump (1.38 mol C m-2 yr-1 and 1.2 mol C m-2 yr-1, respectively), thus play a large role in the carbon cycle (Thomas et al. 2005; Hjalmarsson et al., 2010). The Skagerrak acts as a main depositional basin for about half of the refractory carbon produced in the North Sea and for a high amount of labile organic matter imported with waters of the near-bottom current from the Danish coast or caused by intense algal blooms (Boon et al., 1999). Input of nutrients largely regulate food webs, which makes nutrients of great economic importance for the coastal areas worldwide (Micheli, 1999; FAO, 2016) and in the Skagerrak nutrients are particularly important for the Nordic fisheries (Hop et al., 1992; Iversen et al., 2002; Olsen et al., 2004; Skogen et al., 2007). Fisheries

and aquaculture sectors of the Skagerrak, commercially valuable for the Scandinavian nations, also make a relevant contribution to a growing global demand for food, which will largely rely on coastal regions to host a major part of food production in the future (e.g. FAO, 2013). Effects of increased primary production range from positive impacts on growth, size and reproduction of fish and shellfish populations to disruptive alterations in the food webs, thus yielding or reducing the profit rates of fisheries (Hop et al., 1992; Micheli, 1999; Iversen et al., 2002; Olsen et al., 2004, 2005; Breitburg et al., 2009; FAO, 2016). Negative changes in trophic levels of the Skagerrak ecosystem have been attributed to overfishing (Cardinale and Svedäng, 2004), however, ongoing studies alert about adverse impact of increased nutrient inputs driving heavy phytoplankton blooms and eutrophication in the region (e.g. Baden et al., 1990; Aure et al., 1996; Breitburg et al., 2009). Eutrophication causes high demand and depletion of oxygen in the bottom waters, which affect species diversity, morphology and population growth, and forces organisms to migrate (Rosenberg et al., 1990; Conley, 2009). Thus, to better understand the ongoing and possible future productivity changes and associated environmental effects, more historical studies are needed.'

Reviewer 2: Also the actual relevance of the outcome of the study to the problems raised in the introduction is not clear and should be made clear in the conclusions. It is an interesting and relevant study, but please make it clear to the reader, too.

Response: After revising discussion part we will make sure that the conclusions addresses the research questions raised in the introduction.

Minor comments: Reviewer 2: Latin grammar rules means that the name of the species should be "Melonis barleeanus", not "Melonis barleeanum". Response: We have changed everywhere to Melonis barleeanus.

Reviewer 2: Are you sure that Cassiculina neoteretis is present in the material? If yes, this could indicate an high influx of deep Atlantic or even Nordic Sea water. "

Response: We sincerely thank the reviewer for underlining this feature. The identification of C. laevigata vs C. neoteretis in the Skagerrak material was done based on studies by Mackensen and Hald (1988) and Seidenkrantz (1995). C. neoteretis was also found there by other studies e.g. Erbs-Hansen et al (2011). Since relative abundance of C. neoteretis exceeds 5% only in two samples in each of the cores we considered it as a rather rare species. However, we have now had a closer look at the changes in the C. neoteretis records We observed a peak of high relative abundance of C. neoteretis between ∼ 1750 − 1800 years which correlates well with a drop in BWT and $\delta$18O at the same time, indicating colder bottom water temperature. Therefore, it is possible that this is an indication of a higher influx of deep Atlantic Water or even Nordic Sea waters (Mackensen and Hald 1988, Seidenkrantz 1995). We will add this information in the revised manuscript.

Reviewer 2: Page 4, lune 25-30: It is not quite clear from the description of the water masses, which ones are surface waters, intermediate waters and bottom waters. One for one water mass is the depth in the water column provided. As the depth of outflowing/inflowing waters is very important, this must be made clear. It also needs to be specified very clearly, which water mass sweep the actual study sites in the deep Norwegian Trench. Response: See the comment above regarding water mass clarification.

Reviewer 2: Page 5, line 12-15: It should be pointed out specifically that both cores are taken from the deep Norwegian Trench. Response: We have added a sentence to the material and method section as follows: "Both cores are taken below sill depth within the deep waters of the Norwegian Trench."

Reviewer 2: Material: please provide a short, overall description of the sediment in the two cores. We have added a following sentence: "Both cores consist of mostly soft organic-rich clay and show no significant changes in grain size and lithology throughout the records."

Reviewer 2: Factor loadings are provided and used in the discussions and presented

well in the final figure of proxy comparison. However, in order to evaluate the results of factor analyses, the authors should consider actually plotting them vs age in a diagram comparing them to the faunal data. Response: We have modified the figure with foraminiferal fauna according to the comment.

[Figure]

[Figure]

Fig. 3

**Fig. 1.**

[Figure]

Fig. 2.

Fig.5

[Figure]

**Fig. 3.**

[Figure]

[Figure]

**Fig. 4.**

Figure capture

Fig. 2 Age model description will be presented in the new version of the manuscript.

Fig. 3 Foraminiferal assemblages including dominant and accessory benthic species for both cores (EMB046/10-4GC and EMB046/20-3GC), and CABFAC results. Absolute abundance is shown as grey filed, while relative abundance as black curve with symbols. The absolute abundance of total benthic foraminifera is a sum of all species: agglutinated (Agglut.) and calcareous (Calc.). Dashed lines divide record into 3 periods of most pronounced palaeoproductivity changes, which are discussed in the text.

Fig. 4 (A) Oxygen stable isotopes ($\delta^{18}$O); (B) the Mg/Ca-derived bottom water temperature (Mg/Ca-derived BWT); (C) the stable oxygen composition ($\delta$w), of both studied cores against age. The errors bands represent uncertainties of the records. (D) Figures include data from depths at which data of all proxies ($\delta$18O, Mg/Ca-derived BWT, $\delta$w) are available (symbols). The curves correspond to 5-points running average. Grey box indicates range of instrumentally recorded temperatures of the time period between 2009 and 1924 years from the area between 57°.17–58°´N and 8°–9°.79´E at 300 – 340 dbar (ICES 2010).

Fig. 5 Comparison of absolute abundances of planktonic foraminifera, total organic carbon (TOC), stable carbon isotope ($\delta$13C), absolute abundance of palaeoproductivity fauna, CABFAC results, absolute abundance of total benthic foraminifera, Mg/Ca-derived bottom water temperature (Mg/Ca-derived BWT), oxygen stable isotope ($\delta$18O) between two studied sediment cores EMB046/10-4GC (green curves) and EMB046/20-3GC (black curves). The thicker curves correspond to 5-points running average. The errors bands represent uncertainties of the records. Dashed lines divide record into 3 periods of most pronounced palaeoproductivity changes, which are discussed in the text.

Fig. 6 Comparison of absolute abundances of planktonic foraminifera, total organic carbon (TOC), stable carbon isotope ($\delta$13C), absolute abundance of palaeoproductivity fauna, CABFAC results, absolute abundance of total benthic foraminifera, the stable oxygen composition ($\delta$w), Mg/Ca-derived bottom water temperature (Mg/Ca-derived BWT), oxygen stable isotope ($\delta$18O) between two studied sediment cores EMB046/10-4GC (green curves) and EMB046/20-3GC (black curves). The reconstructions of the NAO index. The thicker curves correspond to 5-points running average. The errors bands represent uncertainties of the records. Dashed lines divide record into 3 periods of most pronounced palaeoproductivity changes, which are discussed in the text.

**Fig. 5.**

---

## Author Response (AR2)

**Technical correction on "Coastal primary productivity changes over the last millennium: a case study from the Skagerrak (North Sea)" by Anna Binczewska et al.**
**Referee2: Marit-Solveig Seidenkrantz**

The authors have done a thorough job at answering reviewers' comments, and I only have a few more or less technical comments.
Response: We appreciate final technical comments from: Marit-Solveig Seidenkrantz. Responses to the review are below.

Reviewer 2: Core description: please also add colour of the sediment, and descript potential structures (laminated, homogeneous, bioturbation etc).
Response: We have added core description, as follow: 'Both cores consist of mostly homogeneous soft organic-rich clay, have olive-grey colour and show no significant changes in grain size and lithology throughout the studied intervals.'

Reviewer 2: Fig 1: map: use different colours for surface vs subsurface waters.
Response: We have modified Fig. 1 according to the comment.

Reviewer 2: Are CTDs available for the sites or near-by locations. If yes, please show these.
Response: The CTDs are available from near-by locations. We have added new Figure 6 with CTD data as supplementary.

Reviewer 2: Please add link to/doi number of data on PANGAEA, when available.
Response: We have added doi.

Reviewer 2: Correct the English of the text, especially in the new sections. There are currently some rather funny misspellings, and odd/incorrect word order.
Response: The manuscript has been tweaked and the language has been tightened up.